# communications
# engineering

# Nonlinear noise spectrum measurement using a probability-maintained noise power ratio method

Tong Ye [1], Xiaofei Su[1], Ke Zhang [1], Chengwu Yang[1], Jingnan Li [1], Yangyang Fan[1], Hisao Nakashima[2], Takeshi Hoshida[2] & Zhenning Tao [1]✉

Nonlinear distortion (noise) limits many communication systems, demanding a means of estimating system performance via device nonlinear characteristics. The noise power ratio (NPR) method which was proposed in 1971 solves this problem for systems with Gaussian stimuli or with special nonlinearity, but practical and accurate methods for many communication systems with non-Gaussian stimuli are rare. Here we propose a probability-maintained (PM) NPR method to accurately measure the spectrum of nonlinear noise via a spectrum analyzer in non-specific systems, including systems with non-Gaussian stimuli. Using an equivalent additive noise model in which the spectrum of equivalent nonlinear noise is the measurement result of PM NPRs, nonlinear system performance could be estimated with an error of 0.5 dB. Further, we find that zero-mean Chi-square noise, instead of Gaussian noise, should be selected for large memory and low-order nonlinear systems. Our method is verified in seven different scenarios with various nonlinear mechanisms and communication applications.

[1] Fujitsu Research and Development Center, No.8 Jianguomenwai Ave, Chaoyang District, Beijing, China. [2] Fujitsu Limited., 1-1 Kamikodanaka 4-Chome, Nakahara-ku, Kawasaki 211-8588, Japan. ✉email: taozn@fujitsu.com

Communication systems are the foundation of our modern society, and nonlinear distortion in these systems is a widespread problem that has piqued the interest of researchers for many years. In optical communication systems, nonlinearities exist in electrical and optoelectronic devices, including transceivers[1–4], lasers[5–7], and optical fibers[8–10]. In wireless communication systems, power amplifiers in base station[11–15], elements in microwave transceivers[16,17], and traveling wave tube amplifiers in satellite channels[18–20] have nonlinear effects. In solid-state circuits, radio frequency high-speed transistors[21–23] always demonstrate nonlinear characteristics. Digital-to-analog converters (DACs) and analog-to-digital converters (ADCs) have integral and differential nonlinearities[24]. Since linear distortions have been well overcome by linear equalizers, nonlinear distortions still severely limit communication system performance. Thus, it is critical to understand the limitation. In other words, it is required to estimate nonlinear communication system performance in terms of bit error ratio (BER) from nonlinear characteristics of devices used in communication systems.

Various nonlinear specifications, such as the total harmonic distortion, are used to characterize nonlinear distortions[25,26]. However, nonlinear distortions not only depend on a nonlinear device but also input signal characteristics[3,27–30], such as power, spectrum, and probability distribution function (PDF). Considering that an input signal in actual communication typically differs from the test signal used in the measurement of conventional nonlinear specifications, it is difficult to estimate actual system performance from conventional nonlinear specifications[26,31–33].

One accurate method for estimating nonlinear system performance is orthogonal decomposition[32–34], whose test stimulus is the signal in actual communication. The output signal $y(t)$ of a nonlinear system is decomposed into the correlated part $y_c(t)$ and orthogonal part $y_o(t)$[35]. The correlated part is the best linear approximation[36] of the output signal, and the orthogonal part is the rest. Thus, the orthogonal item cannot be expressed by an input signal with any linear processing. It could be considered the main contributor of nonlinearity[27,35–39]. The difficulty in measuring is a disadvantage of the orthogonal decomposition method[33]. Coherent comparison of input and output signals in the time or frequency domain is necessary. In addition, as output and correlated components are significantly larger than the orthogonal component, the calculation of $y_o(t)$ may have a large error once linear approximation or time synchronization has a small error. As a result, nonlinearity estimation solutions[17,40–42] based on the orthogonal component always need precise synchronization and high-accuracy calculation[32,33]. Actually, if we can obtain an accurate output signal, system performance can be computed directly.

The noise power ratio (NPR) method[43], which was first proposed in the 1970s, is a simpler solution. In this method, the input signal eliminates a narrow frequency bandwidth component. The output spectrum regrowth at the notch is considered nonlinear noise because no new frequency component is generated by linear effect, as shown in Fig. 1a. With digital signal processing (DSP), adding a deep notch to the input spectrum is not difficult. NPR solutions only compare input and output spectra, which can be measured separately or even in separated locations. This facilitates the measurement[29]. In addition, the two spectra can be measured in different domains, for example, in the electrical and optical domains, respectively. This is essential for devices with different input and output types, for example, optoelectronic devices. However, for non-specific systems, the spectrum regrowth at the notched frequency will be larger than actual nonlinear noise if the input signal is not Gaussian[44–46]. The difference could be as large as 8 dB in some cases[46]. Here, the

"non-specific" means that the system nonlinearity contains more than even-order components[32,33]. Thus, the NPR method is severely restricted to cases with Gaussian stimuli or the systems with special nonlinearity[32,33]. It cannot be used for non-specific communication systems based on pulse amplitude modulation (PAM) and quadrature amplitude modulation (QAM) signals[45,47], which is the main reason why the NPR method has faded out over the years.

In this article, we propose a probability-maintained (PM) notch at the symbol-domain to generate a test signal with a frequency notch while retaining as many of the characteristics of an actual communication signal as possible. The measured spectrum regrowth at the notched frequency turns out to be the correct evaluation of nonlinear noise using such a test signal. In addition to the improvement in accuracy, the advantage of easy measurement in NPR is preserved in the proposed method. We also propose an equivalent additive noise model that can accurately estimate nonlinear system performance by using NPR measurement results as the spectrum of equivalent nonlinear noise. In addition, we find that the distribution of equivalent nonlinear noise should be zero-mean Chi-square instead of Gaussian for nonlinear systems with large memory and low-order. By experiments and simulations, the proposed method is extensively verified in seven scenarios with different nonlinear mechanisms. The verified cases include representative Volterra model, electrical driver, vertical cavity surface-emitting laser (VCSEL), distributed feedback (DFB) laser, electrical DAC, optical coherent transmitter, and optical fiber nonlinear effect. The open issue and future challenges are also discussed in the "Supplementary Discussion" section of supplementary information. This method accurately measures nonlinear distortions for non-Gaussian stimuli in non-specific systems by using simple spectrum analysis. This revives the 50-year-old NPR method for nonlinear communication systems.

## Results and discussion

**Failure analysis of conventional noise power ratio method.** To facilitate repeating the proposed method, we use numerical simulation to illustrate the method, as shown in Fig. 1b. The nonlinear system is a 3rd-order Volterra model, and the input signal is PAM8. Volterra series is a model for nonlinear behavior[22] that can capture memory effects, whose coefficients are listed in Supplementary Data 2. A 3rd-order Volterra model with a time-domain input $x(t)$ and output $y(t)$ can be denoted as Eq. (1), where $h_{k,l,m}^{(1,2,3)}$ are the series coefficients, and $N_{1,2,3}$ represent the memory length of 3 orders.

$$y(t) = \sum_{k=-\frac{N_1-1}{2}}^{\frac{N_1-1}{2}} h_k^{(1)} x(t-k) + \sum_{k=-\frac{N_2-1}{2}}^{\frac{N_2-1}{2}} \sum_{l=-\frac{N_2-1}{2}}^{\frac{N_2-1}{2}} h_{k,l}^{(2)} x(t-k) x(t-l)$$

$$+ \sum_{k=-\frac{N_3-1}{2}}^{\frac{N_3-1}{2}} \sum_{l=-\frac{N_3-1}{2}}^{\frac{N_3-1}{2}} \sum_{m=-\frac{N_3-1}{2}}^{\frac{N_3-1}{2}} h_{k,l,m}^{(3)} x(t-k) x(t-l) x(t-m)$$

$$(1)$$

We first analyze the failure of the conventional NPR method. After processing using the Volterra model and Fast-Fourier Transform (FFT) algorithm, the output spectrum regrowth is calculated by comparing the notch depth of receiver (Rx) and transmitter (Tx) signals. Meanwhile, the actual nonlinear noise is the component orthogonal to the Tx signal. Orthogonal decomposition is used to calculate the Tx-correlated signal and the orthogonal part. As shown in Fig. 1ci, the spectra of orthogonal components, whose stimuli are PAM8 signals with and without a notch, differ, especially at the notched frequency. The orthogonal spectrum of notched PAM8 signals has a hump.

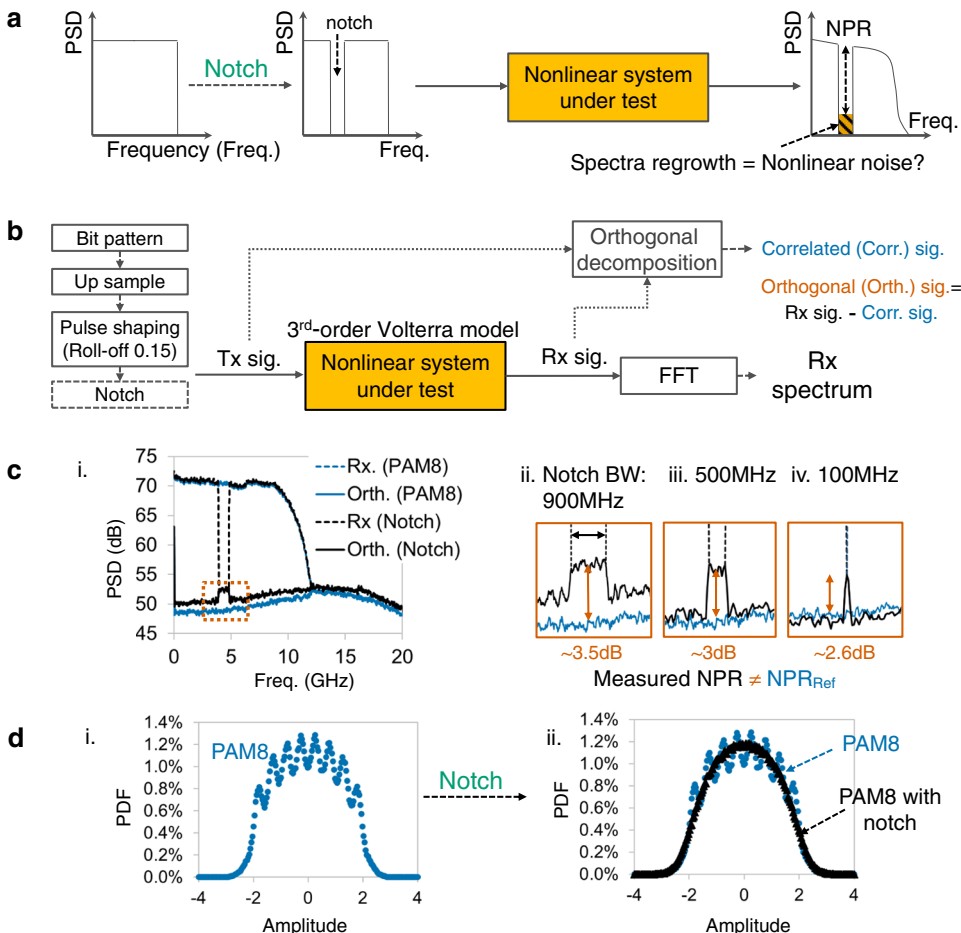

**Fig. 1 Failure analysis of conventional Noise Power Ratio (NPR) method. a** Schematic of conventional Noise Power Ratio (NPR) method. PSD: Power spectrum density. Freq.: Frequency. **b** Simulation setup of NPR analysis by orthogonal decomposition and spectrum comparison. Tx: Transmitter. Rx: Receiver. Sig.: Signal. Corr.: Correlated. Orth.: Orthogonal. FFT: Fast-Fourier Transform. **c** i. With PAM8 as input signal and 3rd-order Volterra as nonlinearity, the measured NPR by simply notching the input signal is not equal to the reference (ref) NPR calculated by orthogonal decomposition. Detailed figures of hump with 900 MHz (ii), 500 MHz (iii), and 100 MHz (iv) notch bandwidth show the difference. PAM: Pulse amplitude modulation. BW: Bandwidth. Blue and black dashed lines: Rx of PAM8 and notched signal. Blue and black polylines: orthogonal component of PAM8 and notched signal. **d** Conventional NPR method fails because the notching process changes the probability distribution function (PDF) of the PAM8 signal. i. PDF of PAM8 signal. ii. Blue circles and black triangles represent the PDFs of PAM8 and PAM8 with notch.

As a result, the conventional NPR measured by simply notching the input signal is not equal to the reference NPR at a notched frequency, which is the power ratio of the PAM8 orthogonal component and Rx signal. The difference is ~3 dB. Figure 1cii, ciii, civ show the detailed spectra of the hump with three different notch bandwidths. Even if the notch frequency bandwidth reduces to as narrow as 100 MHz, the measured NPR remains incorrect. The essential reason for this failure is that the notching process changes the PDF of the Tx signal, as shown in Fig. 1d. As nonlinear characteristics strongly depend on the input signal's PDF, the simple notch process changes the nonlinear activity and makes the conventional NPR method fail for non-Gaussian stimuli.

**Probability-maintained NPR method**. To keep the spectrum notch and not to change the activity of a nonlinear system simultaneously, we first propose the PM NPR method, which uses the PM notch signal as the input signal. Taking PAM8 as an example, Fig. 2ai shows the flow diagram of generating the PM notch symbol sequence. It consists of three steps iteratively. The initial input signal could be any kind of random signal, such as a white Gaussian random signal. We also generate reference samples with the desired PDF and the same sample number as the input signal. As shown in

Fig. 2aii, the first step is "construct PDF," which guarantees the desired PDF and does not significantly change the spectrum[29]. In this step, the samples of the input signal are replaced by the reference samples while maintaining the order. For example, if the maximum sample of the input signal is found at time index 19, it is replaced by the maximum sample in the reference samples. Since the standard PAM8 has sorting ambiguity, we use the diffused PAM8, which is PAM8, along with a small random value, as the reference samples. After that, this new sequence has an identical PDF with reference samples while maintaining a similar PSD. Spectrum adjustment is the 2nd step, as shown in Fig. 2aiii, where the spectrum of a signal after PDF construction is divided into hundreds of resolution blocks. Here, the resolution block has the same concept as the resolution bandwidth of a spectrum. The total power of each resolution block should match the corresponding block power of the PAM8 signal in actual communication. Besides, we add a random perturbation within each resolution block to escape from the local optimum. In the perturbation process, frequency components within each resolution block are multiplied by a set of random values. The 3rd step is notching the signal spectrum. Notching one or more frequency slots are both permitted. Steps 2 and 3 generate the desired spectrum with a notch, but the

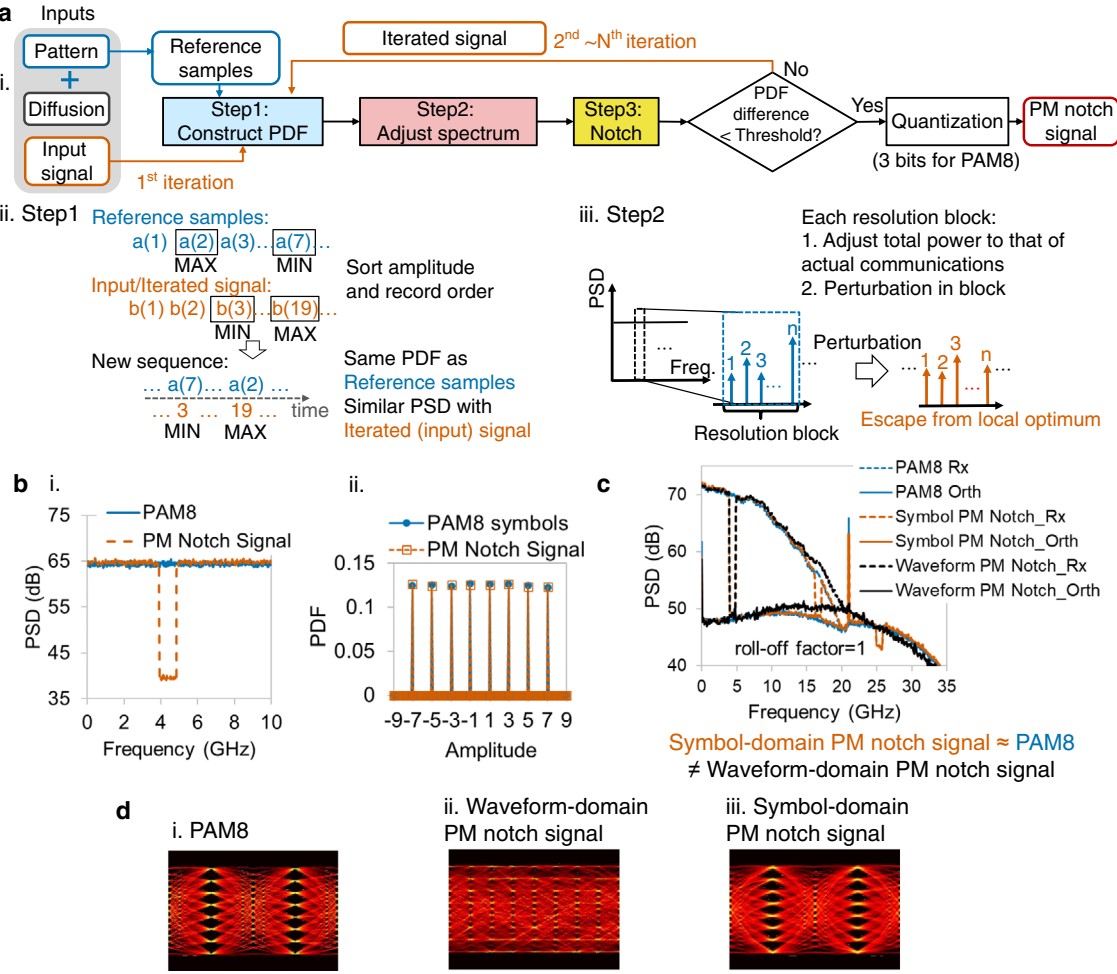

**Fig. 2 Probability-Maintained NPR method. a** Flow diagram of generating Probability-Maintained (PM) notch symbol sequence (i), and the detailed flows of "step 1: construct PDF" (ii) and "Step 2: adjust spectrum" (iii). MAX: Maximum. MIN: Minimum. **b** The PSD (except notched frequency) (i) and PDF (ii) of PM notch symbol sequence (vermillion dashed lines) are as same as PAM8 (blue polylines). **c** Notching at the symbol-domain is necessary. Waveform-domain PM notch signal does not maintain the intra-symbol structure as PAM8, which results in a difference between its Rx and orthogonal spectra and those of PAM8 in roll-off 1 case. Blue, vermillion, and black dashed lines: Rx of PAM8, symbol-domain PM notch signal, and waveform-domain PM notch signal. Blue, vermillion, and black polylines: orthogonal component of PAM8, symbol-domain PM notch signal, and waveform-domain PM notch signal. **d** The difference also can be seen in the eye diagrams of PAM8 (i), waveform-domain PM notch signal (ii), and symbol-domain PM notch signal (iii).

PDF may deviate from the desired one. The difference between the PDF of the generated signal sequence and that of reference samples are calculated as $PDF\ difference = \frac{1}{2}\sum_N |PDF_{gen}(i) - PDF_{ref}(i)|$. N is the total number of bins in calculating the PDFs, and $i$ is the bin index. If their PDF difference is larger than the threshold, return to step1. Otherwise, the iteration stops, and the signal is quantized to PAM8. Finally, the PAM8 signal with a frequency notch and the same PDF as the ideal PAM8 is generated, which is called the PM notch signal. The spectrum and PDF of the PM notch signal are shown in Fig. 2bi, ii, respectively. The PM notch signal is essentially a specially designed PAM8 sequence, and it can be transmitted in any communication system. Thus, the proposed method is appropriate for most types of communication systems. The detailed illustration for generating PM notch symbol sequence is in the "Supplementary Methods" section of supplementary information. The Supplementary Data 3 provides a series of PM notch symbol sequence with six different notch frequencies.

Unlike the 50-year-old conventional NPR method, the proposed PM notch signal is notched in the symbol-domain instead of the waveform-domain. "Waveform-domain notch" means the notch process occurs after digital pulse shaping, whereas "symbol-domain notch" means the notch process occurs before

digital pulse shaping. In the 50-year-old conventional NPR method, the notch process occurs before nonlinear device, and it is waveform-domain notch. An example to illustrate the necessity of symbol-domain notching is shown in Fig. 2c. The waveform-domain PM notch signal has the same PDF and notched PSD with the PAM8 waveform after Nyquist root-raised-cosine pulse shaping with a roll-off factor of 1. After processing with the same Volterra nonlinear model and orthogonal decomposition, the Rx or orthogonal spectrum of the waveform PM notch signal differs from that of the PAM8 stimulus, whereas the outputs of the symbol PM notch match well with both spectra. The reason is that waveform-domain notching destroys the intra-symbol structure. The eye diagrams of the three stimuli clearly show this in Fig. 2d: unlike the other two signals, the waveform-domain PM notch signal (Fig. 2dii) has a closed eye diagram. Thus, all the notch process including both PM notch and simple notch, are symbol-domain notch in following.

Simulation verification for Probability-Maintained NPR method is shown in Fig. 3. Figure 3a shows simulation results of the PM notch signal in the same 3rd-order Volterra model as Fig. 1b. The orthogonal spectrum of the PM notch stimulus has the same profile as that of the PAM8 stimulus so that its NPR will

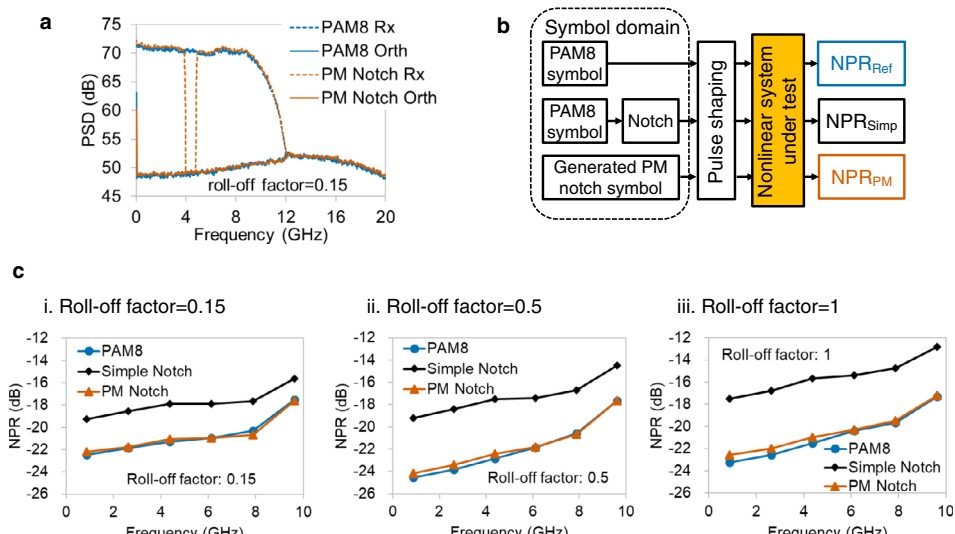

**Fig. 3 Simulation verification for Probability-Maintained NPR method. a** With 3rd-order Volterra as nonlinearity, the spectrum of orthogonal items is equal to that of PAM8 when a PM notch signal is used as a stimulus. Blue and vermillion dashed lines: Rx of PAM8 and PM notch signal. Blue and vermillion polylines: orthogonal component of PAM8 and PM notch signal. **b** Three types of NPRs with different symbol-domain stimuli will be compared in the following, including $NPR_{Ref}$ by PAM8, $NPR_{Simp}$ by symbol-domain simple notch signal, and $NPR_{PM}$ by symbol-domain PM notch signal. Ref: Reference. Simp: Simple. **c** For roll-off factors 0.15 (i), 0.5 (ii), and 1 (iii), the $NPR_{PM}$ estimated by symbol-domain PM notch signals (vermillion triangles) are close to $NPR_{ref}$ (blue circles), whereas $NPR_{Simp}$ (black diamonds) has a large error. Baud rate: 21 Gbaud.

be close to the reference NPR. For unbiased comparison, the simple notch is also implemented in the symbol-domain for the conventional NPR method. Figure 3b defines 3 NPRs, which will be compared below. $NPR_{Simp}$ and $NPR_{PM}$ are the notch depths directly measured using the output spectra of the simple and PM notch signals. $NPR_{Ref}$ regards the orthogonal item of PAM8 as "noise," which represents the actual level of nonlinearity.

In Fig. 3ci, ii, iii, the $NPR_{Ref}$, $NPR_{Simp}$, and $NPR_{PM}$ are simulated with three different roll-off factors of Nyquist root-raised cosine pulse shaping. Six PM notch symbol sequences are generated, and each signal has one notched frequency slot. Figure 3c shows that for all roll-off factors, the PM notch NPR method can measure accurate NPRs with a root-mean-square error (RMSE) within 0.4 dB, whereas $NPR_{Simp}$ has a large error. It is verified that the PM NPR method can estimate the orthogonal power spectrum of nonlinear systems accurately, irrespective of pulse shaping. To this end, we will focus on cases with a roll-off factor of 0.15 in the following to avoid repetition.

**Equivalent additive noise model.** Nonlinear output can be regarded as the summation of the linear correlated part and orthogonal item[35]. Thus, a nonlinear system can be approximated using an equivalent linear model and equivalent additive noise[12,36,39], and the additive noise is independent of the specific input signal bit pattern (Fig. 4ai). As shown in Fig. 4aii, the linear model is the input signal passing through a filter, which is obtained by comparing the input and output spectra of the nonlinear system. The equivalent additive noise is a random noise to approximate nonlinear distortions. This is named the equivalent additive noise model. To focus on nonlinear distortion, we assume that the receiver has an ideal linear equalizer. The equivalent additive noise model is a linear system so that the system performance could be estimated easily.

One choice on equivalent additive noise is the time-domain orthogonal item[32,34], for instance, the orthogonal signal of another input bit pattern or the orthogonal item with sample shifting (Fig. 4aiii). In this so-called "same-orthogonal model," the additive noise maintains the same PDF, PSD, intra-symbol structure, and

joint probability density of the noise at neighboring symbols with an actual orthogonal signal. It could be considered the performance upper limit of the equivalent additive noise model. The system performance is typically specified by Q factor[48,49], which can be calculated by BER, as shown in Eq. (2). Figure 4bi, bii show Q factors estimated using the same-orthogonal model under the conditions that the additive white Gaussian noise (AWGN) in the communication channel has signal-to-noise ratios (SNRs) of 21 and 30 dB. Both another bit pattern additive noise and orthogonal item shifting 2000 samples can achieve Q factors similar to actual system performance. Although the same-orthogonal model has high estimation accuracy, the measurement is difficult, and it cannot be used practically[33].

$$Q\ factor = 20\log_{10}\left[\sqrt{2}erfc^{-1}(2BER)\right]dB, \qquad (2)$$

Considering that the PM notch signal can accurately measure the spectrum of the orthogonal item, a simple and realistic solution, named "same-spectrum model," is proposed, whose additive noise has the same PSD as the actual orthogonal signal. As shown in Fig. 4aiv, the same-spectrum additive noise can be constructed by passing white X-distributed noise through the NPR filter. Here, X can be Gaussian, our proposed zero-mean Chi-square, proposed negative zero-mean Chi-square, and so on. Zero-mean Chi-square is the PDF of a Chi-square distributed random variable with 1 degree of freedom minus its mean value. A PDF example is shown in Fig. 4av. Since Chi-square is an asymmetric distribution, the negative zero-mean Chi-square that equals the mean value minus Chi-square distribution is also investigated. The mean of the Gaussian or Chi-square noise is zero since "zero-mean noise" is widely assumed in nonlinear system performance analysis[36,39]. In real implementation, this assumption is guaranteed by alternative current coupling in the receiver. The inner-band part of the NPR filter is the linear interpolation of measured NPRs, whereas the outer-band part is the outer-band spectrum of the Rx signal. The NPR filters determine the power and covariance of additive noise. Same-spectrum models with various noise distributions are investigated in both medium and small AWGN cases, as shown in Fig. 4ci, ii,

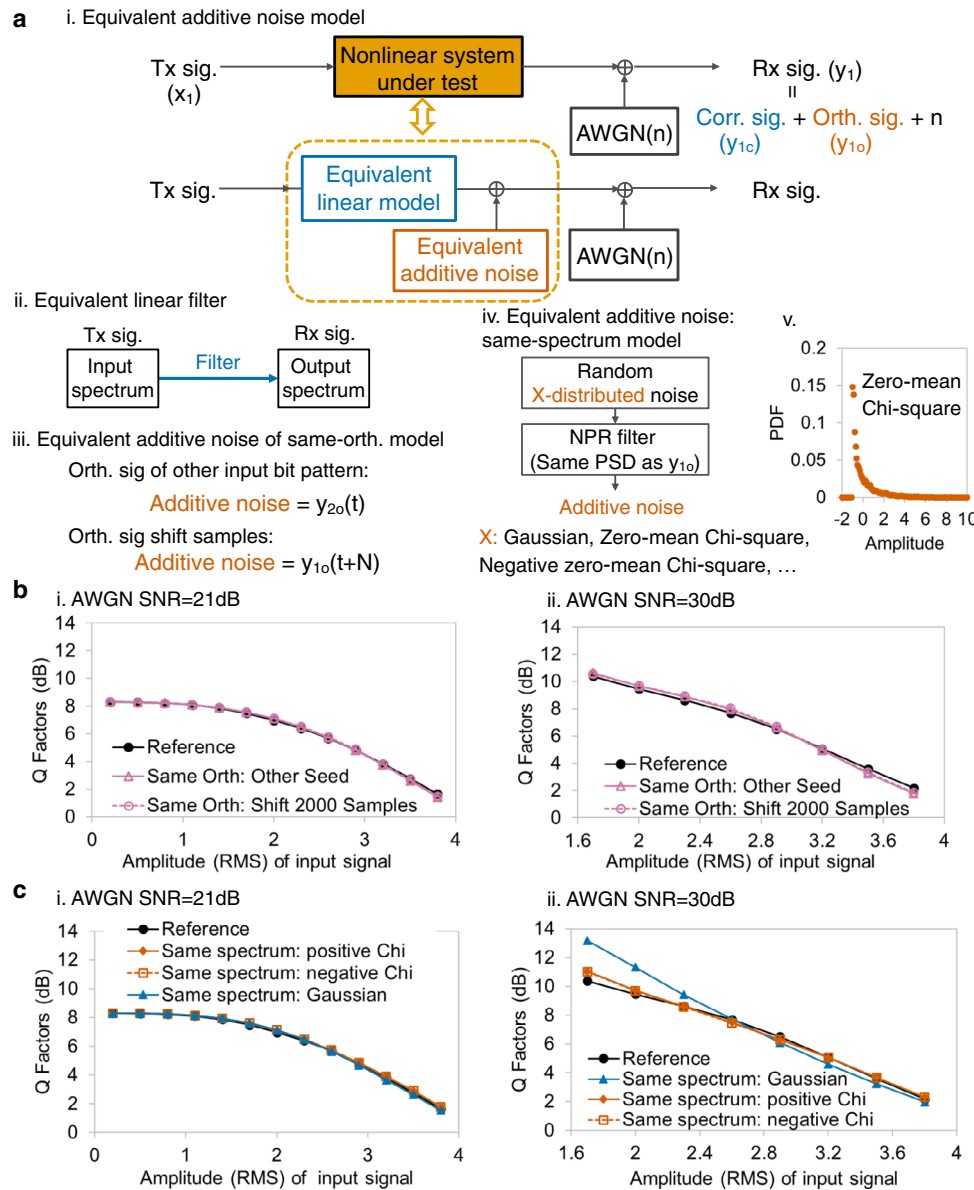

**Fig. 4 Equivalent additive noise model. a** Nonlinear system can be approximated by the sum of equivalent linear model and equivalent additive noise (i). The linear model is a filter calculated by comparing input and output spectra (ii), and the equivalent additive noise can be an orthogonal item (iii) or same-spectrum noise (iv). v. PDF of zero-mean Chi-square noise. AWGN: Additive white Gaussian noise. **b** For both medium (i) and small (ii) AWGN cases, the same-orthogonal model (reddish purple line with hollow triangles and dashed line with reddish purple hollow circles) has system performance similar to the actual system (black line with circles). The horizontal axis is the root-mean-square (RMS) amplitude of the input signal, which affects system nonlinearity. SNR: Signal-to-noise ratio. **c** Q factors estimated by same-spectrum noise are close to reference Q at medium AWGN case (i), whereas the difference among various noise distributions stands out with 30 dB AWGN SNR (ii). For the 3rd-order Volterra model, the same-spectrum model with zero-mean Chi-square noise (polyline with vermillion diamonds and dashed line with vermillion hollow squares) has the smallest estimation error. Blue line with triangles: same-spectrum model with Gaussian noise.

respectively. With identical data processing flow, the reference Q is obtained using the actual time-domain output, and Q factors of models are obtained from the constructed linear signals and additive noise. For a medium AWGN case with 21 dB SNR, the same-spectrum model Q is close to reference Q, and various noise distributions have a small difference. This result means that for the case with medium channel AWGN, the PSD of nonlinear noise is sufficient for estimating the system Q performance. For negligible AWGN cases, where nonlinear distortion dominates, the difference among random noise distributions stands out. Interestingly, the Gaussian-distributed additive noise model has a larger estimation error than zero-mean Chi-square noise,

including positive and negative ones. An intuitive but not strict explanation is illustrated in the following.

An instanced nonlinear system is a linear filter plus 3rd-order memoryless nonlinearity. Supposing the transmitted symbol at index 0 is $s_0$, the nonlinear system output at index 0 is

$$y_0 = c_1(s_0 + \delta) + c_2(s_0 + \delta)^2 + c_3(s_0 + \delta)^3, \quad (3)$$

where $\delta$ represents the inter-symbol interference (ISI) caused by the finite impulse response (FIR) before nonlinearity. It could be assumed as Gaussian if FIR has a large memory. The nonlinear

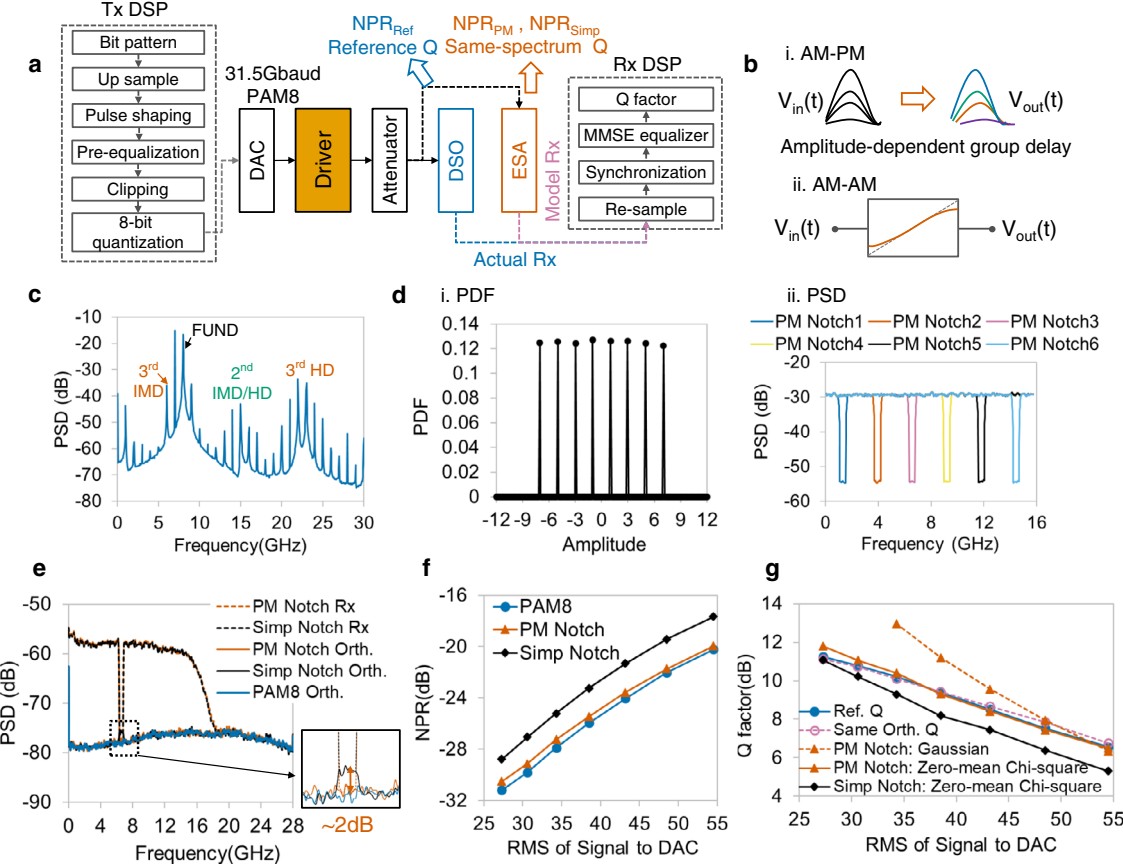

**Fig. 5 Experimental verification using driver. a** Experimental setup and digital signal processing (DSP) flows of Tx and Rx. DAC: Digital-to-analog converter. DSO: Digital storage oscilloscope, used to obtain the time-domain waveform of an output signal, which has a similar function as an analog-to-digital converter. ESA: Electrical spectrum analyzer. MMSE equalizer: Minimum mean square error equalizer. **b** Schematic of driver's nonlinearity. i. AM-PM nonlinearity. ii. AM-AM nonlinearity. AM: Amplitude modulation. PM: Phase modulation. **c** Output spectrum of two-tone stimulus shows that the 3rd-order nonlinearity is dominant in driver. FUND: Fundamental wave. HD: Harmonic distortion. IMD: Intermodulation distortion. **d** PDF (i) and PSD (ii) of PM notch signals. Six PM notch signals with different notched frequency are marked as blue, vermillion, reddish purple, yellow, black, and sky-blue lines in PSD figure. **e** Spectrum analysis of PM and simple notch signals. The bottom of PM notch output is close to the orthogonal spectrum of PAM8, whereas the notch depth of the simple notch signal is smaller than reference NPR. Vermillion and black dashed lines: Rx of PM notch signal and simple notch signal. Vermillion, black, and blue polylines: orthogonal component of PM notch signal, simple notch signal, and PAM8. **f** Comparison between reference NPRs (blue circles), PM NPRs (vermillion triangles), and conventional NPRs (black diamonds). PM notch NPRs are close to reference ones under different input RMS value. **g** Estimation performance of equivalent additive noise model. Same-spectrum model constructed using PM NPRs, and zero-mean Chi-square random noise (polyline with vermillion triangles) can accurately estimate the system Q factors (blue circles). Dashed line with reddish purple hollow circles: same-orthogonal model. Dashed line with vermillion triangles: same-spectrum model with PM notch NPRs and Gaussian-distributed noise. Black diamonds: same-spectrum model with simple notch NPRs and zero-mean Chi-square noise.

system output $y_0$ can be rewritten as

$$y_0 = \left(c_1 s_0 + c_2 s_0{}^2 + c_3 s_0{}^3\right) + \left(c_1 + 2c_2 s_0 + 3c_3 s_0{}^2\right)\delta \\ + \left(c_2 + 3c_3 s_0\right)\delta^2 + c_3\delta^3, \quad (4)$$

The term in the first bracket is deterministic. The 2nd term can be removed by the receiver linear equalizer. The 3rd term has a Chi-square distribution with 1 degree of freedom, but the mean value can be removed by alternative current coupling in the receiver. The final 4th is small and negligible. Thus, the equivalent noise is assumed a zero-mean Chi-square distribution.

These results show that the same-spectrum model can estimate system performance using a linear filter and PM NPRs, and the distribution of random additive noise should be chosen carefully.

In this section, taking the Volterra model as an example, we analyze why the conventional NPR method fails, and propose a PM notch NPR at the symbol-domain to obtain the actual spectrum of the orthogonal item. Then, the same-spectrum model with PM NPRs can be used to accurately estimate nonlinear

system performance. In the following, we will verify the proposed estimation method in various nonlinear scenarios. To highlight the nonlinearity and distinguish between various random noise distributions, we will exclude the Rx-side AWGN in the following experiments and simulations.

**Driver nonlinearity.** The PM NPR method and equivalent additive noise model are verified using an electrical driver used in the optical coherent transmitter, and the experimental setup is shown in Fig. 5a. As shown in Fig. 5b, the nonlinearity of the driver includes the so-called amplitude modulation-phase modulation (AM-PM) and amplitude modulation-amplitude modulation (AM-AM) effect, which are amplitude-dependent group delay and nonlinear relationship between input and output amplitude[23,50]. By applying two frequency tones as input signal, the characteristics of driver nonlinearity are investigated. As shown in Fig. 5c, the output spectrum of the two tones has larger 3rd harmonic distortions (HD) and intermodulation distortions

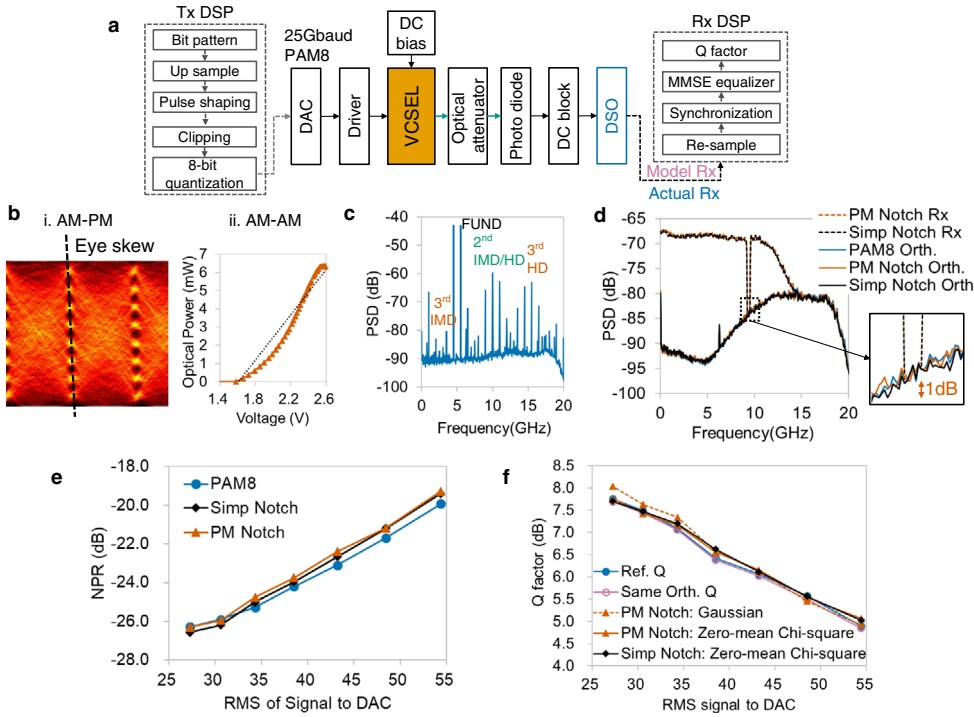

**Fig. 6 Experimental verification of VCSEL nonlinearity. a** Experimental setup and DSP flows of Tx and Rx. VCSEL: Vertical cavity surface-emitting laser. DC: Direct current. **b** i. AM-PM nonlinearity: eye skew under nonlinear conditions. ii. AM-AM nonlinearity: measured voltage-optical power relationship. **c** Output spectrum of two-tone stimulus shows that the 2nd-order nonlinearity is dominant in VCSEL. **d** Spectrum analysis of PM and simple notch signals. The bottoms of both PM and simple notch outputs are close to the orthogonal spectrum of PAM8. Vermillion and black dashed lines: Rx of PM notch signal and simple notch signal. Vermillion, black, and blue polylines: orthogonal component of PM notch signal, simple notch signal, and PAM8. **e** Comparison among reference NPRs (blue circles), PM NPRs (vermillion triangles), and simple NPRs (black diamonds). Both PM and simple notch NPRs are close to reference ones under different input RMSs. **f** Estimation performance of equivalent additive noise model. Blue circles: reference Q. Dashed line with reddish purple hollow circles: same-orthogonal model. Dashed line with vermillion triangles: same-spectrum model with PM notch NPRs and Gaussian-distributed noise. Polyline with vermillion triangles: same-spectrum model with PM notch NPRs and zero-mean Chi-square noise. Black diamonds: same-spectrum model with simple notch NPRs and zero-mean Chi-square noise.

(IMD) than 2nd-order HD, which means that the 3rd-order nonlinearity is dominant in the driver.

The PDF and PSD of the PM notch signal are shown in Fig. 5di, ii, where 8 amplitude levels are uniformly distributed, and 6 notched frequencies correspond to 6 PM notch signals. Figure 5e shows the output spectra of PM and simple notch signals, and the depth of the notched frequency is the measured NPR. By orthogonal decomposition, the orthogonal spectra of 3 stimuli, including reference signal PAM8, PM notch signal, and simple notch signal, are obtained for comparison. The Rx notch bottom and orthogonal spectrum of the PM notch signal coincide with the PAM8 orthogonal item, whereas the orthogonal spectrum of the simple notch stimulus has an obvious hump. The estimation error of notched signals is investigated under different input powers. As shown in Fig. 5f, the RMSE of the PM notch signal is <0.55 dB, and that of the simple notch is 2.02 dB.

The performance of the equivalent additive noise model is also verified experimentally, as shown in Fig. 5g. For the same-orthogonal model using the sample-shifted orthogonal item as additive noise, the Q factors are almost the same as reference Q. For the additive noise generated by PM NPRs, because of the driver's 3rd-order nonlinearity, the estimated Q is close to reference Q when the random noise is zero-mean Chi-square distributed, whereas the Gaussian noise-estimated Q has a large error with small input power. For the model using conventional NPRs, even if the noise is zero-mean Chi-square, the estimated Q error is still large. This verifies that the driver's nonlinear performance can be accurately estimated using the PM NPR

method and the proposed same-spectrum equivalent noise model, with a small RMSE of 0.26 dB on the Q factor.

**VCSEL nonlinearity.** In optical intensity modulation and direct detection (IM-DD) systems, VCSEL is a widely used component with non-negligible nonlinearity. The experimental setup and DSP flow for an 850 nm multi-model VCSEL are displayed in Fig. 6a. The nonlinearity results mainly from VCSEL by setting other components linearly. The AM-PM nonlinearity of VCSEL is typically presented as eye skew[51], as shown in Fig. 6bi. The eye diagram of the PAM8 signal slopes under nonlinear conditions. As VCSEL converts an electrical signal into optical power, its AM-AM nonlinearity can be observed by the voltage-optical power relationship. Compared with the dashed trend line in Fig. 6bii, the measured relationship (in vermillion) is nonlinear. By analyzing the output spectrum of two tones in Fig. 6c, VCSEL nonlinearity is 2nd-order dominant, which is different from that of the driver.

Figure 6d shows the output spectra of PM and simple notch signals, and both have notch bottoms close to the spectrum of orthogonal item. For different input powers, both PM and simple notch signals can obtain NPRs with a small error of ~0.5 dB, as shown in Fig. 6e. The results means that the simple notch method may be feasible for 2nd-order nonlinearity dominant system, which is also experimentally verified in IM-DD VCSEL system with PAM4 inputs[52].

The Q factors estimated using equivalent models are shown in Fig. 6f. Q values of the same-orthogonal model are similar to

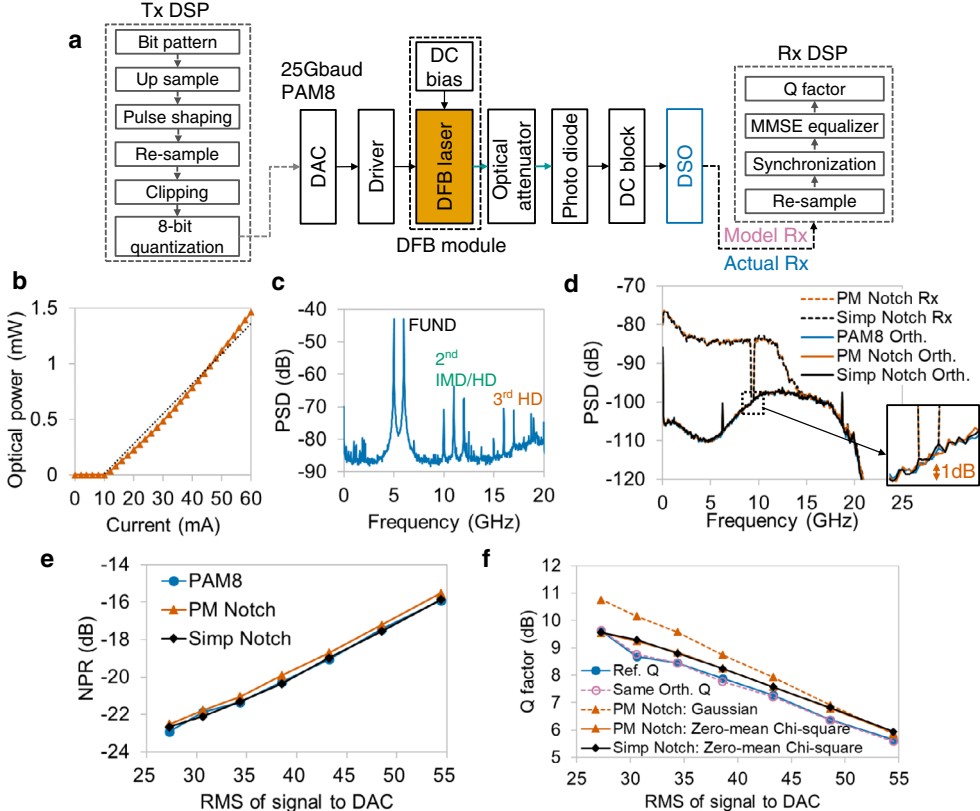

**Fig. 7 Experimental verification of DFB laser nonlinearity. a** Experimental setup and DSP flows of Tx and Rx. DFB: Distributed feedback. **b** Measured current-optical power relationship shows nonlinearity in DFB laser. **c** Output spectrum of two-tone stimulus shows that the 2nd-order nonlinearity is dominant in DFB laser. **d** Spectrum analysis of PM and simple notch signals. The bottoms of both PM and simple notch outputs are close to the orthogonal spectrum of PAM8. Vermillion and black dashed lines: Rx of PM notch signal and simple notch signal. Vermillion, black, and blue polylines: orthogonal component of PM notch signal, simple notch signal, and PAM8. **e** Comparison among reference NPRs (blue circles), PM NPRs (vermillion triangles), and simple NPRs (black diamonds). Both PM notch NPRs and simple notch NPRs are close to reference ones under different input RMSs. **f** Estimation performance of equivalent additive noise model. Blue circles: reference Q. Dashed line with reddish purple hollow circles: same-orthogonal model. Dashed line with vermillion triangles: same-spectrum model with PM notch NPRs and Gaussian-distributed noise. Polyline with vermillion triangles: same-spectrum model with PM notch NPRs and zero-mean Chi-square noise. Black diamonds: same-spectrum model with simple notch NPRs and zero-mean Chi-square noise.

reference Q. For the same-spectrum Q estimated by PM NPRs and zero-mean Chi-square noise, the RMSE is ~0.1 dB. The results of the PM NPR and conventional NPR methods are similar because both have small errors in nonlinear noise spectrum estimation. As the nonlinearity of VCSEL is mainly of 2nd order, the Chi-square additive noise slightly outperforms Gaussian-distributed noise.

**DFB laser nonlinearity**. A DFB laser is another widely used directly modulated laser in optical IM-DD systems, whose working wavelength is 1550 nm. The experimental setup and DSP for the Tx and Rx sides are shown in Fig. 7a. Similar to VCSEL, the nonlinearity of the DFB laser can be observed in the current-optical power relationship, which is listed in Fig. 7b. As shown in Fig. 7c, the 2nd-order dominant nonlinearity is also verified by the two-tone spectrum.

The spectra of PM notch and simple notch outputs are plotted in Fig. 7d. All curves, including Rx spectra and orthogonal components of notched signals and PAM8, are overlapped at the notched frequency. As the DFB laser is also a 2nd-order nonlinearity dominant system, the accuracy of PM and simple notch-measured NPRs is confirmed under various input powers with small estimation errors, as shown in Fig. 7e.

Considering the 2nd-order nonlinearity of DFB laser, it is reasonable that the estimation error of the same-spectrum model with zero-mean Chi-square noise is smaller than that with Gaussian noise. The estimation results are plotted in Fig. 7f. The RMSE of the same-orthogonal model and the same-spectrum model is 0.06 and 0.37 dB, respectively, when the random noise is zero-mean Chi-square distributed. The results illustrate the feasibility of the equivalent same-spectrum model with Chi-square noise in IM-DD systems.

**Electrical DAC nonlinearity**. The nonlinearity in electrical DACs is investigated by using a high-speed 4-bit DAC (SHF Communication Technologies AG, 612 A). The experimental setup and DSP for the Tx and Rx sides are shown in Fig. 8a. Unlike an ordinary DAC whose input digital signal is stored in the memory, the input signals of this DAC are four analog binary nonreturn-to-zero (NRZ) signals (Fig. 8bi). These four analog NRZs are generated using a four-branch DAC and four drivers, and each NRZ represents the signal of 1 bit. The output of the 4-bit DAC is synchronously merged by these four analog inputs with adjustable amplitude levels. In an ideal case, the amplitude levels of the 4 bit DAC are proportional, and the ratio of voltages should be 1:2:4:8. By deviating the amplitude level from the ideal value, different DAC nonlinearity is emulated, as shown in Fig. 8bii. The tested

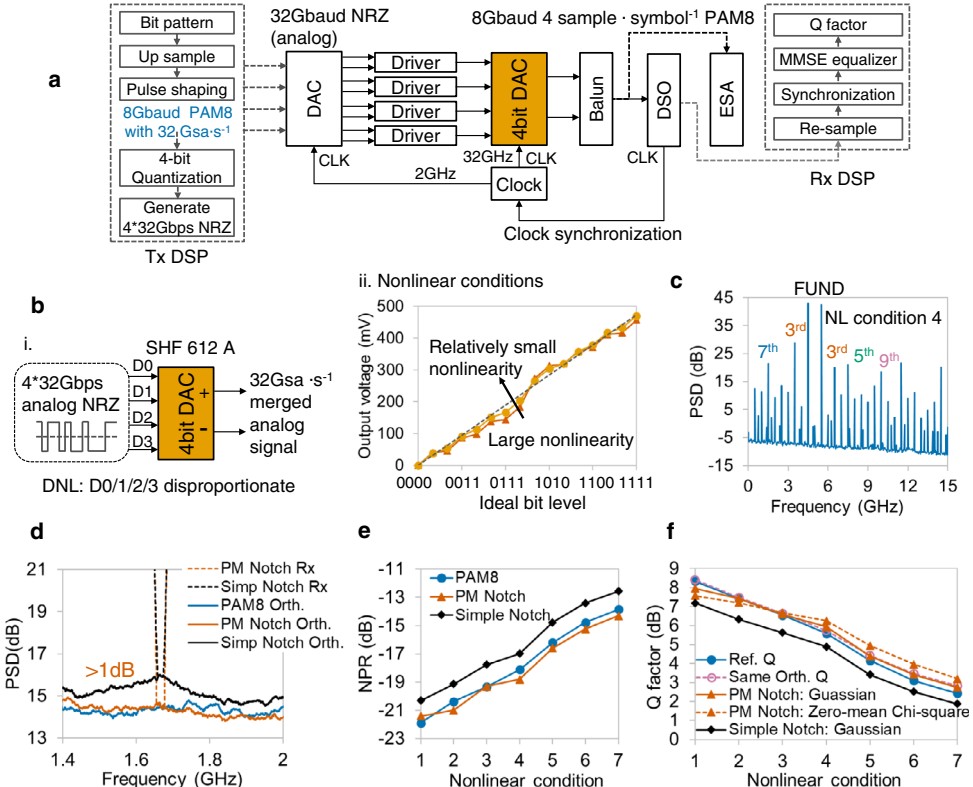

**Fig. 8 Experimental verification of electrical DAC nonlinearity. a** Experimental setup and DSP flows of Tx and Rx. NRZ: Nonreturn-to-zero. Gsa•s⁻¹: G samples per second. Gbps: G bits per second. Balun: Balance-unbalance converter. CLK: Clock. **b** i. DAC nonlinearity results from the disproportionate amplitude levels of 4 bits. SHF: SHF Communication Technologies AG. DNL: Differential nonlinearity. ii. The corresponding output voltages of the DAC output positive port for ideal bit levels 0000-1111 show disproportional values. Vermillion triangles and yellow circles represent large and relatively small nonlinearities. **c** Output spectrum of two-tone stimulus shows that the nonlinearity of DAC is odd-order dominant nonlinearity, and high-order nonlinearities, such as 5th, 7th, and 9th order, exist. NL: Nonlinear. **d** Spectrum analysis of PM and simple notch signals. The bottom of PM notch output (vermillion) is close to the orthogonal spectrum of PAM8 (blue), whereas that of the simple notch (black) has a large gap. **e** Comparison among reference NPRs (blue), PM NPRs (vermillion), and simple NPRs (black). For all nonlinear conditions, the PM NPR method can accurately measure the nonlinear noise, and simple NPRs have a large error of ~1.5 dB. **f** Estimation performance of equivalent additive noise model. Same-spectrum model with PM NPRs, and Gaussian-distributed noise (polyline with vermillion triangles) has Q factors similar to reference system (blue circles). Dashed line with reddish purple hollow circles: same-orthogonal model. Dashed line with vermillion triangles: same-spectrum model with PM notch NPRs and zero-mean Chi-square noise. Black diamonds: same-spectrum model with simple notch NPRs and Gaussian noise.

two-tone spectrum in Fig. 8c shows that the nonlinearity is odd-order dominant nonlinearity, and it has non-negligible high-order nonlinearities, including 5th, 7th, and 9th order nonlinearities.

The output spectra of PM and simple notch signals are shown in Fig. 8d. The notched bottom of the PM notch output is close to the orthogonal part of the PAM8 reference. For the output spectrum of the simple notch, the bottom of the notched slot has a gap larger than 1 dB compared with the actual nonlinear noise. Nonlinear conditions 1–7 which represent DAC nonlinearity from small to large errors, are investigated by comparing the measured NPRs and actual nonlinear noise. Figure 8e shows that the PM NPR method is verified to be accurate, with an estimation error less than 0.5 dB.

Figure 8f illustrates the estimation results of the equivalent models. The same-orthogonal model and same-spectrum model with PM NPRs and Gaussian-distributed noise have small estimation RMSEs of 0.24 and 0.28 dB, respectively. For the same-spectrum model with Chi-square random noise, the estimated Q factors have an obvious bias against reference Q. The reason why Gaussian noise is superior may be the high order of nonlinearity. Compared with the 2nd/3rd-order nonlinear system, the orthogonal component contains more nonlinear items when higher-order nonlinearities exist. That combination makes the distribution of orthogonal signals close to Gaussian. Thus, for

nonlinear devices with high-order nonlinear distortions, the PM notch method and the same-spectrum model with Gaussian noise can accurately evaluate communication system performance.

**Nonlinearity in optical coherent transmitter.** Figure 9a shows the experimental setup of an optical coherent back-to-back (B2B) transmission system. PM and simple NPRs are measured using an optical spectrum analyzer (OSA), and the reference ones are obtained by decomposing data from a digital storage oscilloscope (DSO). The DSO is used to obtain the time-domain waveform of output signals, whose function is similar to that of ADC. Only one polarization is used to transfer signal. The unmodulated direct current light of the other polarization is used as a pilot to remove the frequency offset and phase noise because the orthogonal decomposition assumes zero phase noise. As shown in Fig. 9b, nonlinearity is caused by the high-bandwidth coherent driver modulator (HB-CDM); the nonlinear mechanisms of the coherent modulator badly influence system performance. The nonlinearity of HB-CDM contains a nonlinear phase-voltage relationship[2,4] (Fig. 9bi) and sine nonlinearity attributed to the Mach-Zehnder modulator structure[53] (Fig. 9bii). According to the working function of a coherent IQ modulator, I and Q tributaries have real-value nonlinearity, respectively, and the

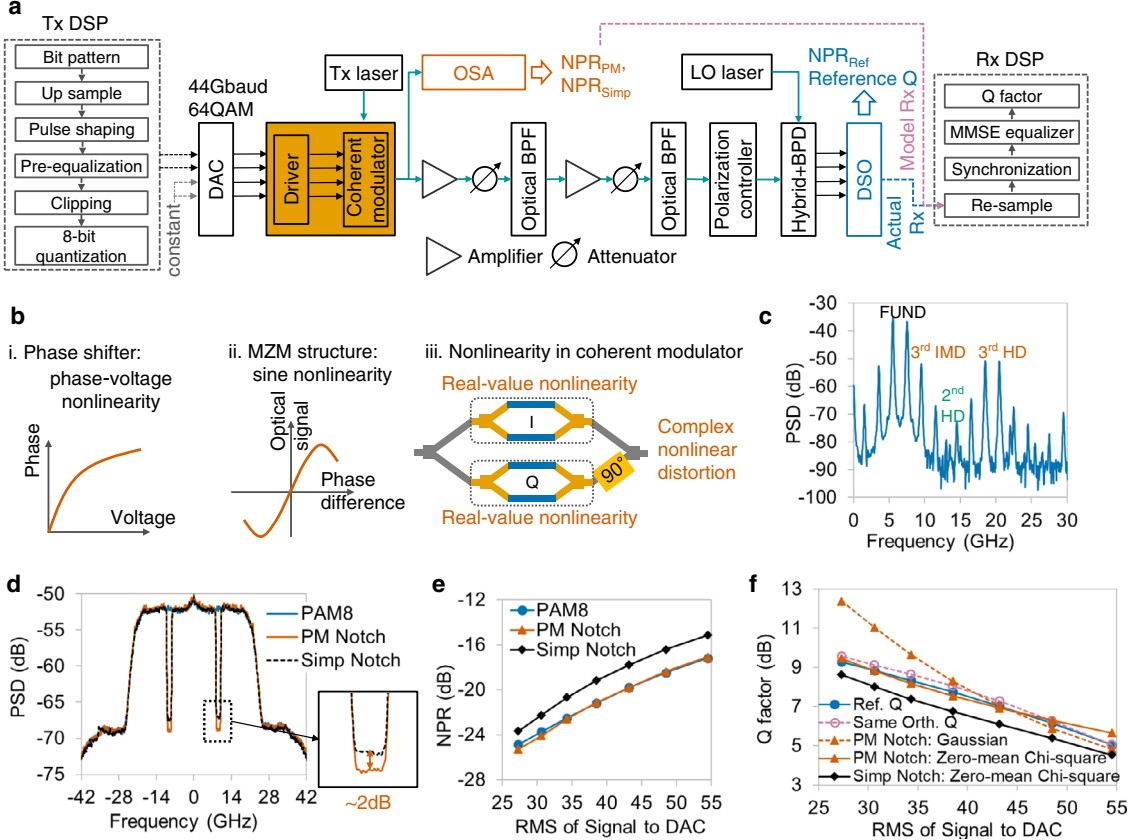

**Fig. 9 Experimental verification of optical coherent transmitter nonlinearity. a** Experimental setup and DSP flows of Tx and Rx. QAM: Quadrature amplitude modulation. OSA: Optical spectrum analyzer. BPF: Bandpass filter. BPD: Balanced photodiode. LO: Local oscillator. **b** The nonlinearity mechanism of coherent modulator: Phase-voltage nonlinearity (i) and sine nonlinearity (ii). MZM: Mach-Zehnder modulator. The optical nonlinear noise is complex-valued, which is combined by two real-value nonlinearities (iii). **c** Output spectrum of two-tone stimulus shows that the nonlinearity of optical coherent transmitter is 3rd-order nonlinearity dominant. **d** Optical spectrum analysis of PM (vermillion) and simple notch (black) signals. The bottom of PM notch output is deeper than that of simple notch signal. Blue: spectrum of PAM8. **e** Comparison among reference NPRs (blue circles), PM NPRs (vermillion triangles), and simple NPRs (black diamonds). For all input powers, the PM NPR method can accurately measure the nonlinear noise, but NPRs measured using the conventional NPR method have large errors. **f** Estimation performance of equivalent additive noise model. Same-spectrum model with PM NPRs and zero-mean Chi-square-distributed noise (polyline with vermillion triangles) has Q factors similar to reference system (blue circles) due to 3rd-order dominant nonlinearity. Dashed line with reddish purple hollow circles: same-orthogonal model. Dashed line with vermillion triangles: same-spectrum model with PM notch NPRs and Gaussian noise. Black diamonds: same-spectrum model with simple notch NPRs and zero-mean Chi-square noise.

complex-value nonlinearity signal is generated by combining I and Q signals with a 90° phase shifter (Fig. 9biii). Thus, NPRs measured using the OSA are the ratios between optical complex-value noise and signals, different from the aforementioned real-value nonlinear scenarios, including the driver, VCSEL, DFB laser, and electrical DAC. The two-tone spectrum in Fig. 9c shows that the nonlinearity of the HB-CDM is 3rd-order dominant.

The spectra of PM and simple notch signals are measured in the optical domain, as plotted in Fig. 9d, the notch depth of the former is ~2 dB deeper than that of the latter. By further comparing the measured PM and conventional NPRs with reference ones under various input powers, the accuracy and superiority of the PM NPR method can be verified, as shown in Fig. 9e.

The estimation results of the complex-value equivalent model are shown in Fig. 9f, and Q factors of the same-orthogonal and same-spectrum models with PM NPRs and zero-mean Chi-square noise have small RMSEs of 0.25 and 0.26 dB, respectively. As the conventional NPRs are incorrect, the same-spectrum model using these NPRs has a large error in terms of Q performance. Owing to the nonlinearity in the optical transmitter being 3rd-order dominant, the zero-mean Chi-square noise is significantly more accurate than Gaussian-distributed noise.

**Nonlinearity in optical fiber**. The fiber transmission, including the linear chromatic dispersion[8] (CD) effect, where different frequency components have different delays, and the nonlinear Kerr effect, where the fiber refractive index changes with the intensity of optical signals[9], is investigated by simulation. Here, simulation instead of experiment is selected by intention because the laser phase noise, the amplifier spontaneous emission noise, and the polarization uncertainty in the real transmission cause errors in the orthogonal decomposition. All components are set linearly, except for the single-mode fiber (SMF). The lasers of Tx and Rx have no frequency offset or phase noise, as shown in Fig. 10a. The mechanism of fiber nonlinearity can be modeled using the split-step Fourier method[54–56], as shown in Fig. 10b. The SMF is split into several slices, and each slice has CD and nonlinear phase modulation—the Kerr effect. Both CD and Kerr effects act on complex-value signals, and the detailed formulas of these complex-value nonlinearities are shown in Fig. 10b. Figure 10c shows that the 3rd-order IMDs and HDs of two-tone signal are large, meaning that 3rd-order nonlinearity is dominant in fiber.

For the fiber nonlinearity, both the $1 \times 150$ km single-span transmission and $10 \times 80$ km multi-span transmission are investigated. At first, the notch bandwidth is 600 MHz. For

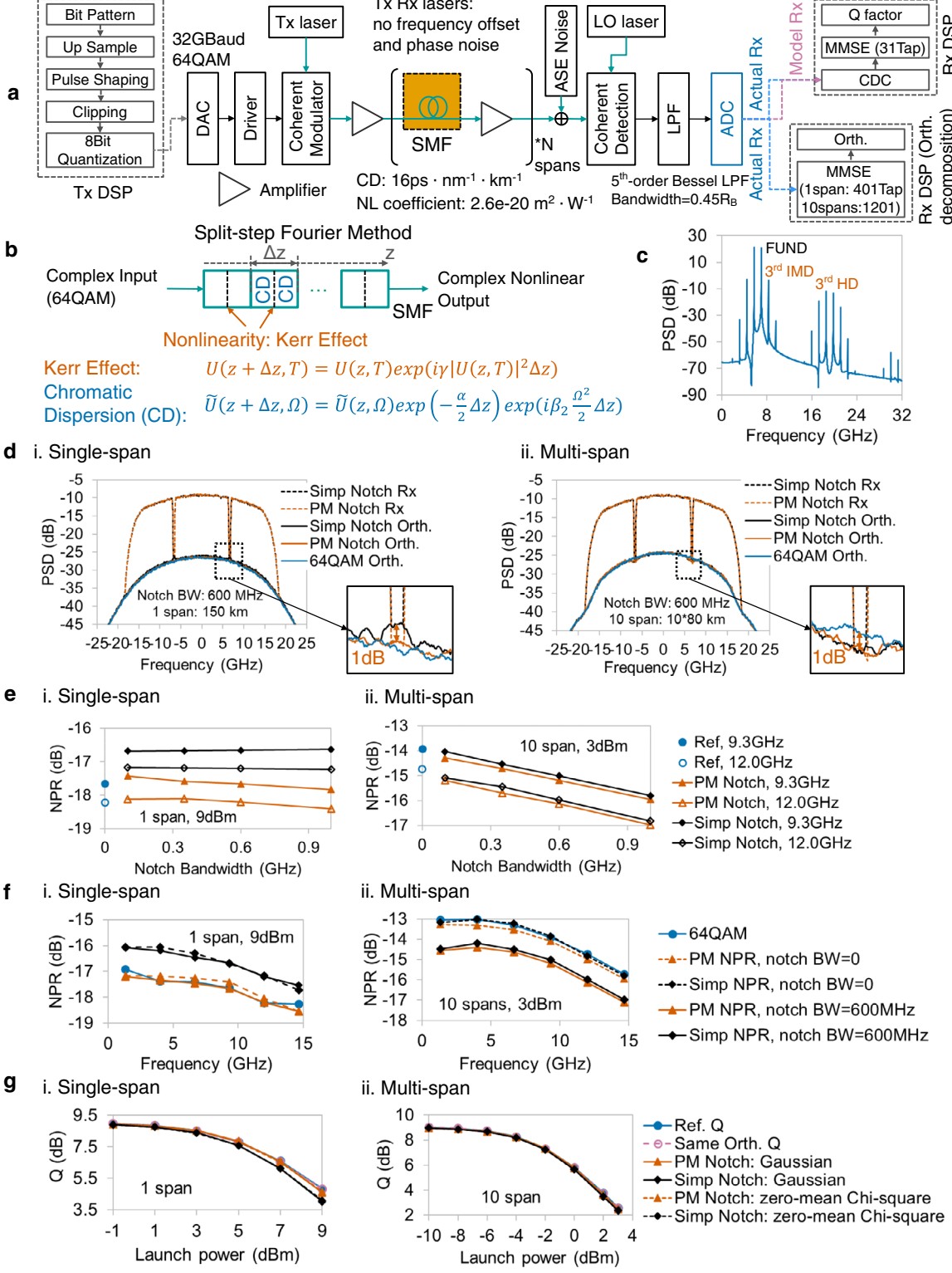

the single-span case (Fig. 10di), the hump can be observed at the notched frequency of simple notch output spectrum, which leads to a 1 dB error for simple NPR method. The PM notch method estimates the similar value of orthogonal item of 64QAM. It is an expected phenomenon in 3rd-order dominant nonlinearity. However, for the multi-span case in Fig. 10dii, the results of both the PM NPR method and the simple NPR method do not agree with the reference NPR noted as "64QAM Orth." This is quite unusual.

To find the reason, the impact of notch bandwidth is investigated. Figure 10e shows the results of different notch bandwidth and the extrapolations at two notched frequency points in single- and multi-span transmissions. For single-span transmission (Fig. 10ei), the NPR values are almost unchanged along with the notch bandwidth, while this phenomenon is different in multi-span transmission (Fig. 10eii). The PM notch signal has a notch in the spectrum, so that it is correlated in time domain. In other words, the PM notch signal has a special order

**Fig. 10 Simulation verification of optical fiber nonlinearity. a** Simulation setup and DSP flows. SMF: Single-mode fiber. CD: Chromatic dispersion. ASE: Amplified spontaneous emission. LPF: Low pass filter. $R_B$: Baud rate. CDC: Chromatic dispersion compensation. **b** The nonlinearity mechanism of split-step Fourier method. Both input and output of fiber nonlinearity are complex-valued. The formula of CD and Kerr effects is denoted using two equations: $U(z,T)$ and $\widetilde{U}(z,\Omega)$ are the time- and frequency-domain optical fields, respectively. Here, $\Delta z$ denotes the step length for split-step Fourier method, $\gamma$ denotes the nonlinear coefficient of Kerr effect, $\alpha$ denotes the loss factor, and $\beta_2$ denotes the group velocity dispersion coefficient. **c** Output spectrum of two-tone stimulus shows that the nonlinearity of fiber is 3rd-order nonlinearity dominant. **d** i. For single-span transmission, the output spectra show that the bottom of PM notch (vermillion) is close to the reference orthogonal item (blue), whereas the simple notch orthogonal item (black) has a hump at notched frequency. ii. In multi-span transmission, the bottoms of two notched signals (vermillion and black) are lower than the orthogonal spectrum (blue) if the notch bandwidth is 600 MHz. **e** i. The NPRs of both PM (vermillion) and simple notch (black) methods in single-span transmission has low correlation with notch bandwidth. ii. For multi-span transmission, extrapolation should be implemented to obtain the NPRs at zero notch bandwidth. **f** i. For single-span transmission, the PM NPR method (vermillion) accurately measures NPRs, and simple NPRs (black) have ~1 dB errors. ii. For multi-span transmission, the extrapolated NPRs of both the PM and simple NPR methods are coincide with the reference NPRs. **g** i. Estimation performance of the same-spectrum model for single-span transmission. Same-spectrum model with PM NPRs (vermillion) rather than simple NPRs (black) has Q factors similar to reference system (blue). ii. By using the extrapolated NPR results, the Q factor can be estimated by same-spectrum model accurately in multi-span transmission. Dashed line with reddish purple hollow circles: same-orthogonal model.

compared with the random signal in the real communication. There is difference between the notched signal and the signal in real communication. This does not make sense in the single-span transmission but makes sense in the multi-span transmission. The possible reason is the long memory of CD in the multi-span transmission. It's natural that the difference turns to zero if the notch bandwidth turns to zero. Thus, the extrapolated NPR at zero notch bandwidth is calculated. Figure 10f compares the actual orthogonal NPRs, measured NPRs of simple notch and PM notch at 600 MHz bandwidth, and the extrapolated NPRs at zero notch bandwidth in both single-span (Fig. 10fi) and multi-span (Fig. 10fii) transmissions. The accuracy of PM NPR method is verified in both single-span and multi-span transmissions with 0.18 dB RMSE. For the simple notch NPR, a 0.97 dB RMSE can be observed in single-span case, but its RMSE in the multi-span transmission is small (0.09 dB).

The reason for above interesting phenomenon is the interaction among the fiber attenuation, the CD, and the nonlinear Kerr effect. Due to the fiber attenuation, the fiber nonlinearity mainly occurs at the beginning of one span, so that the nonlinearity in the single-span transmission is close to the lumped nonlinear system, whereas the nonlinearity in the multi-span transmission is a distributed nonlinear system. The observations in single-span transmission are like other nonlinear cases illustrated in this article. In multi-span transmission, there is almost no difference between simple notch and PM notch because the CD causes large ISI, and the signal PDF turns to Gaussian in the subsequent spans.

The Q performances of single- and multi-span transmissions can be estimated, as shown in Fig. 10gi, gii, respectively. For the single-span case (Fig. 10gi), same-orthogonal model and same-spectrum model with PM NPRs can estimate the Q factor of various launch powers accurately. Same-spectrum model with simple notch NPRs has non-ignorable error. For the multi-span case (Fig. 10gii), all the methods evaluate the system performance accurately because the PDF of the input signal to the nonlinear device is Gaussian. It is also interesting that there is almost no difference between the Q of Gaussian noise and zero-meaning Chi-square noise in Fig. 10g. This is attributed to the CD compensation in the receiver. The CD compensation has large ISI, which turns various noise to Gaussian-distributed noise.

## Conclusions

Estimating nonlinear system performance based on device nonlinear characteristics practically is a fundamental demand in the communication field. However, this is challenging because nonlinear distortions depend not only on the nonlinear device but also on the input signal. Conventional NPRs solve this problem in a few cases, such as when the input signal is Gaussian or when the system

has a special nonlinearity. However, in many communication systems, the input signal is not Gaussian-distributed. In this article, we propose a PM notch at the symbol-domain to generate a test signal for NPR measurement. With this test signal, the spectrum of the orthogonal component and the equivalent nonlinear noise could be practically measured with an RMSE of 0.5 dB, even if the input signal is not Gaussian. This cures the long history headache since the 1970s that the NPR method cannot work for non-Gaussian inputs in non-specific nonlinear communication systems. Along with the same-spectrum model, where the equivalent additive nonlinear noise has the same-spectrum as the measurement result of PM NPR, the Q factor of the nonlinear system could be estimated with an RMSE <0.5 dB. The selection of noise PDF is also discussed, and we show that zero-mean Chi-square instead of Gaussian should be used for large memory and low-order nonlinear communication systems. Only spectrum information is required for PM NPR measurement and same-spectrum model construction. It means that the proposed estimation solution is simple and easy to implement. The proposed solution is verified in many scenarios with different nonlinear mechanisms, including the 3rd-order Volterra model, electrical driver, VCSEL, DFB laser, electrical DAC, optical coherent transmitter, and optical fiber. We believe that the symbol-domain PM notch will revive the 50-year-old NPR method.

## Methods

**NPR calculation**. All PM and simple NPRs are calculated from the output spectra of notched signals. As its definition, an NPR is the power ratio of noise and Rx signal; here, noise is calculated using the power of notch bottom, and the Rx signal power is the linear interpolation of the power at positive and negative frequencies with one notch bandwidth away from the notch center. For each frequency, power is calculated by averaging all points within the calculation bandwidth, which equals 50% of the notch bandwidth.

For PM notch signals, the notch depth is ~25 dB for PAM8 due to 3 bit quantization. Here, we denote this Tx-side NPR as $NPR_{Tx}$. The purpose of NPR methods is to measure the increment/regrowth of notched frequency PSD, so the Tx-side NPR should be removed from Rx-measured results $NPR_{Rx}$. Equation (4) shows the calibration method for PM NPRs. For the conventional NPR method, this issue is insignificant because the notch of a simple notch signal is sufficiently deep to ignore $NPR_{Tx}$.

$$NPR_{PM} = 10\log_{10}\left(10^{\frac{NPR_{Rx}}{10}} - 10^{\frac{NPR_{Tx}}{10}}\right). \tag{4}$$

**Same-spectrum model construction**. A linear filter, nonlinear NPR filter, and random noise with X distribution are required to construct the same-spectrum model. For all the investigated cases, the calculation methods for linear filters and nonlinear NPR filters are the same. The linear filter is obtained in two parts: For the frequency within half of the Baud rate, the filter is calculated by comparing Rx and Tx filter responses. For the frequency outside half of the Baud rate, the filter response is a constant value, which equals the response at half the Baud rate. A nonlinear NPR filter is also obtained by splicing in-band and out-band. The filter response of the in-band part, whose frequency range is below $(1 + roll \text{-} off \, factor) \times Baudrate \times 2^{-1}$, is calculated by linear interpolation of the measured NPRs.

We regard the Rx response outside the $(1 + roll - off\ factor) \times Baudrate \times 2^{-1}$ frequency range as nonlinear filter response. For a 2nd-order dominant nonlinear system, the frequency boundary of NPRs should be lower than $(1 + roll - off\ factor) \times Baudrate \times 2^{-1}$ (13.2 GHz in VCSEL and DFB laser cases) because 2nd-order nonlinearity generates more out-band noise than in-band noise. Thus, the linear and nonlinear filter responses can be obtained by Tx and Rx frequency responses and measured NPRs. According to our experimental results, the low resolution of these filter responses, for example, 150 MHz of OSA in the optical B2B system, is acceptable.

In the real-value equivalent model, the X-distributed real-value noise is generated by a random noise and nonlinear NPR filter, For the complex-value equivalent model in optical coherent transmitter and fiber nonlinearity, the real and imaginary part of the X-distributed complex noise are two different real-value noises with specified PDF.

**Details in simulations and experiments**. Our first simulation case is the 3rd-order real-value Volterra model, whose tap numbers for 3 orders are 101, 11, and 5, respectively. An 84-Gsample·s$^{-1}$ and 21-Gbaud PAM8 is used as a reference signal, with a symbol length of 128k. The PM notch and simple notch signals have the same rate and length as the reference signal, and the notch bandwidth is ~900 MHz. Ideal up sampling and root Nyquist cosine pulse shaping are used to generate Tx inputs, and for Rx DSP, the T/4-space MMSE equalizer has 101 taps. In this case, we add medium and small AWGNs at the receiver side, whose SNRs are 21 and 30 dB. The SNR is defined as the ratio between signal power and in-band noise power, and it does not change with input power.

The signal used in the driver nonlinearity experiment is 112-Gsample·s$^{-1}$ 31.5-Gbaud PAM8 with a symbol length of 73728. The driver gain voltage is set to 2.45 V, and the nonlinearity is adjusted by changing the input power which is illustrated by the RMS of signal to DAC with full swing of $-127$–$127$. The notch bandwidth of PM and simple notch signals is 600 MHz. For Tx DSP, after up sampling and pulse shaping, 0.2% clipping, pre-equalization for DAC, and 8 bit quantization are applied. For Rx-side DSP, a T/2-space MMSE equalizer with 51 taps is used to equalize the four-time averaged Rx signal. Averaging is used to remove the AWGN at the Rx-side. In orthogonal decomposition, a 5.3 bit effective number of bits (ENOB) is considered to avoid high-frequency anomalies.

For the IM-DD VCSEL experiment, the sample rate, and Baud rate for PAM8 are 100 Gsample·s$^{-1}$ and 25 Gbaud. The symbol length is 128k, with a notch bandwidth of 400 MHz for PM and simple notch signals. The bias voltage of VCSEL is set to 2.07 V, and the nonlinearity is changed by various input powers. All measured spectra are from DSO with 16 frames averaging to remove the AWGN. In Tx-side DSP, 0.2% clipping and 8 bit quantization are used. The T/2-space MMSE equalizer in Rx-side DSP has 2501 taps in Q calculation, and the T/4-space MMSE approximation has 5001 taps in orthogonal decomposition. The long tap number is necessary because the reflection in the link should be excluded in nonlinear noise. For VCSEL, the ENOB in consideration is 5.4 bits.

In the DFB laser experiment, the settings of the reference signal, PM notch signal, and simple notch signal are the same as those in the VCSEL experiment. The bias current of the DFB laser is 40 mA. The output signal is 64 frames averaged to focus on the nonlinearity. Tx-side DSP is also the same as that of the VCSEL system, and Rx-side DSP uses 1251-tap T/2-space MMSE equalizer in Q calculation. In orthogonal decomposition, the T/4-space MMSE approximation has 2501 taps. As the DAC used in the DFB laser system is the same as that in the VCSEL experiment, the ENOB used in orthogonal decomposition is also 5.4 bits.

For the electrical DAC case, an 8-Gbaud 4-bit-quantized PAM8 is generated with four samples per symbol. Four analog binary NRZs are input into the electrical DAC, and each NRZ represents the signal of one bit. After merging the four analog NRZs using the 4-bit DAC, the differential output signals of the 4-bit DAC can be combined by a BALUN and measured by ESA and 50-Gsample·s$^{-1}$ DSO without averaging. Here, every notch signal has three notched slots, and the bandwidth of each slot is ~121 MHz. For Rx-side DSP, 201-tap T/2-space MMSE equalizer and approximation are used in Q calculation and orthogonal decomposition. A DAC ENOB of 6 bits is used in the calculation.

In the nonlinearity investigation for optical coherent transmitter, the 44-Gbaud 64-QAM signal with two samples per symbol is used. To measure the nonlinear noise in the optical domain, the real and imaginary part of the complex-value PM and simple notch signal are two different real-value notch signals. The symbol length is 128k, and the notch bandwidth is 2 GHz. The gain of HB-CDM is fixed to 17.6 dB, and the nonlinearity is adjusted by the input power with a step of 1 dB. To remove the Rx-side AWGN, 14 frames of DSO data are averaged as a comparison with OSA spectrum results. Pre-equalization, 0.2% clipping, and 8-bit quantization are used in Tx-side DSP. Here, pre-equalization is used to compensate for the bandwidth-limited response of the DAC board and HB-CDM. The signal I + jQ is loaded on one polarization, and the other polarization is an unmodulated direct current light as an auxiliary channel. Then, the frequency offset and phase noise can be estimated using the unmodulated polarization and removed before using the adaptive equalizer (AEQ) in Rx-side DSP. The AEQ used in this case is a 2 × 2 real-value MMSE equalizer in T/2-space with 51 taps. The DAC ENOB is 4.8 bits, which is used in orthogonal decomposition.

Fiber nonlinearity is investigated by simulation. A 32-Gbaud 64-QAM signal is transferred with 16-time oversampling. The notch bandwidth for PM and simple notch signals is 600 MHz. The fiber transmission effect includes CD and Kerr nonlinear effect, whose coefficients are set to 16 ps·nm$^{-1}$·km$^{-1}$ and $2.6 \cdot 10^{-20}$·m$^2$·W$^{-1}$. The nonlinearity is adjusted by various launch powers. As the lasers at the Tx-side and in the local oscillator have no frequency offset and phase noise in simulation, Rx-side DSP only needs CD compensation and 31-tap T/2-space MMSE equalizer for Q factor calculation. For orthogonal decomposition, a long tap T/2-space approximation is required to exclude the CD effect from orthogonal items. The tap number depends on the amount of CD; for a single-span system, 401-tap MMSE approximation is sufficient; for a 10-span system, a tap length of 1201 is sufficient.

## Data availability

Necessary data to repeat the proposed method, including the Volterra model and PM notch signal, are submitted as supplementary data. Source data underlying figures are also submitted.

## Code availability

The proposed method could be repeated by standard communication algorithms in common simulation software, such as MATLAB or Python.

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

## Author contributions

T.Y. conceived the idea of PM NPR method, verified the method on electrical DAC, and analyzed the data of all cases. X.S. conceived the idea of notch at symbol domain and verified the method on 3rd-order Volterra model. K.Z. improved the iteration details of PM NPR method, conceived the idea of zero-mean Chi-square noise, and verified the method on driver and optical coherent transmitter. C.Y., J.L., and Y.F. verified the method on VCSEL, DFB laser, and optical fiber, respectively. T.Y. drafted initial paper, Z.T., T.H., and H.N. provided the major revision and all authors contributed to the final version. Z.T. supervised the project and provided guidance to all co-authors.

## Competing interests

The authors declare no competing interests.
