## [Peer Review File · Communications Engineering]

Nonlinear noise spectrum measurement using a probability-maintained noise power ratio methodReviewers' comments:

Reviewer #1 (Remarks to the Author):

This paper studies a method to estimate the end-to-end system performance of nonlinear (NL) communication systems by measuring only the floor of in-band power spectral density (PSD) that arises from NL distortion (or noise) without recovering the transmitted signal through receiving equipment. The idea introduced to make this possible for non-Gaussian modulation systems is interesting and has some novelty. However, there are a number of important weaknesses that prevent this reviewer from recommending acceptance of this paper for publication in Communications Engineering. Please see below for details.

- The NPR method used throughout the paper relies on the fundamental assumption that any NL distortion manifests itself in the form of spectral entities in the signal band. This assumption is the basis for quantifying the sum power of (linear or NL) noise and NL distortion by measuring the floor of the PSD through a spectral null deliberately created within the signal band. However, not all NL distortion (or noise) gives rise to spectral regrowth in the spectral null. A well-known example is the NL interference in fiber-optic communications caused by optical Kerr effect, which is the last of the seven examples of the NL distortion covered in this paper. More specifically, at any infinitesimally narrow band in the PSD (regardless of whether it contains signal or not), the NL interference power caused by Kerr nonlinearity is approximately proportional to the cube of the optical power within that band, so the spectral regrowth due to NL interference is almost imperceptible in spectral nulls. The fact that significant NL interference power can exist in the spectral bands with high optical power even if it is not found in the spectral nulls implies that the NPR method is not useful for quantifying the NL interference if it is caused by Kerr nonlinearity. This shows that there are NL systems that invalidate the fundamental assumption of this paper, so the method of this paper cannot be generally applied to all NL systems. Since the NL system refers to all systems that are not linear, a huge variety of NL systems can exist, but the method presented in this paper seems to prove its usefulness only when the system is expressed as a cubic polynomial.

- The technical description of the methodologies used is difficult to understand or is missing, leaving the readers to guess vaguely from the texts scattered over figures. For example, it is hard to understand Steps 1-3 in Fig. 2 from both text and the figure. What is the definition of the "resolution block"? What is the definition of the "difference between probability density functions"? There doesn't seem to be any principle used in the paper that would make this "difference between probability density functions" decrease over iterations of Steps 1-3 (such as the gradient descent method), but why do the iterations lead to convergence? Why does the "local optimum" occur?

- Although the proposed method creates a desired probability density function (PDF) of signal, the PDF is not made by independent and identically distributed signals but rather by correlated signals. This correlation does not exist in ordinary PAM signals, and can potentially cause artifacts when measuring NL distortion. To see this, note that when ordinary PAM signals are randomly permuted in time, the PSD does not change, but when the signals generated by the proposed method are randomly permuted in time, the PSD is likely to change. This means that the proposed method generates specially ordered signals in time, and since NL distortion can vary depending on the order of signal in time, the effect of the specially created order on the NL distortion should be analyzed.

- The use of a Chi-square distribution in some of the "equivalent additive noise models" has a physical basis, but shifting the Chi-square distribution to have zero mean is arbitrary and lacks a physical basis. In fact, if the exact mathematical model of an NL system is known, the PDF of NL distortion can be accurately determined through analysis or Monte-Carlo simulation in a similar way to the paper's attempt with equations (3)-(4). However, the explanations surrounding equations (3)-(4) seem to be wrong. To be more specific, when a linear equalizer recovers $y_0 = c_1 * s_0 + z$, with z being NL distortion, equation (4) should first be rewritten in this form and the PDF of z should be characterized.

Perhaps the dominant term of z would be product of two Gaussian random variables, whose PDF is a linear combination of two non-central Chi-square PDFs, not a single Chi-square PDF arbitrarily shifted to have a zero mean.

- For the "equivalent additive noise model" to be accurate, it needs an accurate PDF of the NL distortion. This requires identifying the mathematical model of the NL system under test, which is a very difficult task and may outweigh the benefits of the proposed method.

Reviewer #2 (Remarks to the Author):

This paper proposes a probability-maintained (PM) NPR method and an equivalent additive noise model that can accurately estimate nonlinear system performance. The topic addressed in this paper is important and the manuscript is interesting. I have the following suggestions:

1. In the proposed PM NPR method, the spectrum needs to be adjusted, as shown in the Fig. 2a Step 2. An explanation about the 'perturbation in block' is required.
2. The notation 'NPR_{Simp}' represents the simple notch, what is the relation/difference between the simple notch and waveform-domain notch?
3. The equivalent nonlinear noise can be chosen following Gaussian noise or zero-mean Chi-square noise. How to determine the mean & covariance of the Gaussian noise, or the degrees of freedom of the zero-mean Chi-square noise? Please explain this point.

Reviewer #3 (Remarks to the Author):

In my opinion, this paper is very well-written, with explanations that are very easy to understand, and figures that are well-thought-out. It is interesting to read.

However, there is an issue of how interesting and relevant it is. The technique of NPR is motivated by its (signal-processing) simplicity compared to the reference technique in the paper, correlated-orthogonal signal separation. However, the proposed technique PM-NPR is quite complicated signal-processing wise itself, which is clear from the description in the "Probability-maintained NPR method" section. The reference technique is not so complex or difficult to use as it is, and it is clearly better. And it also gives an entire spectrum as output, whereas the proposed PM-NPR only gives a single numerical value.

Further, the technique is quite ad-hoc, and the credibility is lost when it does not work for all cases (e.g. for fiber nonlinearity), and the authors have no good explanation why. The Chi-Square noise distribution is also weakly motivated, really only through simulations and experiments, and it does not always work.

Thus, the paper has a bit of an "unfinished" feeling, the technique is a bit ad-hoc, and it is questionable if it is worth to study it at all, when the reference technique is quite useful directly.

Conclusion: Interesting paper, but it has quite significant shortcomings. At this point, I am inclined to recommend rejection.

Dear reviewers,

Thank you very much for your thorough reviews and comments. Those comments help us in enhancing the article. We have addressed all the comments and revised the manuscript. In particular, we have proven that the proposed method can still be applied to the case of the fiber Kerr effect through extrapolation. Details are shown in following. Our response is marked in blue and all changes to the original manuscript are marked in underline in the revised manuscript. We believe the revised version meets the requirement of *Communications Engineering*.

Yours sincerely,

Zhenning TAO

Reviewer #1 (Remarks to the Author): -----

--

This paper studies a method to estimate the end-to-end system performance of nonlinear (NL) communication systems by measuring only the floor of in-band power spectral density (PSD) that arises from NL distortion (or noise) without recovering the transmitted signal through receiving equipment. The idea introduced to make this possible for non-Gaussian modulation systems is interesting and has some novelty. However, there are a number of important weaknesses that prevent this reviewer from recommending acceptance of this paper for publication in *Communications Engineering*. Please see below for details.

1 - The NPR method used throughout the paper relies on the fundamental assumption that any NL distortion manifests itself in the form of spectral entities in the signal band. This assumption is the basis for quantifying the sum power of (linear or NL) noise and NL distortion by measuring the floor of the PSD through a spectral null deliberately created within the signal band. However, not all NL distortion (or noise) gives rise to spectral regrowth in the spectral null. A well-known example is the NL interference in fiber-optic communications caused by optical Kerr effect, which is the last of the seven examples of the NL distortion covered in this paper. More specifically, at any infinitesimally narrow band in the PSD (regardless of whether it contains signal or not), the NL interference power caused by Kerr nonlinearity is approximately proportional to the cube of the optical power within that band, so the spectral regrowth due to NL interference is almost imperceptible in spectral nulls. The fact that significant NL interference power can exist in the spectral bands with high optical power even if it is not found in the spectral nulls implies that the NPR method is not useful for quantifying the NL interference if it is caused by Kerr nonlinearity. This shows that there are NL systems that invalidate the fundamental assumption of this paper, so the method of this paper cannot be

generally applied to all NL systems. Since the NL system refers to all systems that are not linear, a huge variety of NL systems can exist, but the method presented in this paper seems to prove its usefulness only when the system is expressed as a cubic polynomial.

Thank the reviewer for the comment, in particular the comment of fiber Kerr effect. We have proven that the proposed PM NPR method can still be applied to the case of the fiber Kerr effect. The important trick is to extrapolate the NPR of zero notch bandwidth from the results of multiple notch bandwidths. As shown in Fig. 9g in our previous manuscript [Fig. 9e in the revised manuscript], the PM NPR does not agree with the reference PNR in the case of multi-span transmission. The reason of this disagreement is the 600 MHz notch bandwidth is not small enough in the case of multi-span transmission. Fig. 9g in the revised manuscript shows that the NPR changes along with the notch bandwidth in multi-span transmission. If we use the extrapolation to obtain the NPR of zero notch bandwidth, the result agrees with the reference NPR obtained by orthogonal component decomposition. For the reason of the notch bandwidth dependency, please refer to our response to the 3rd comment. Fig. 9i shows the PM NPR with zero notch bandwidth agrees with the reference NPR, and Fig. 9k shows the estimated system performance agrees with the reference performance. A new observation in Fig. 9g/9i/9k is that the performance of PM NPR and simple NPR are very close in the case of multi-span transmission. The reason is the chromatic dispersion. The PDF of the input signal of the subsequent span is Gaussian because chromatic dispersion causes large ISI. Please refer to Fig. 9f-9k and corresponding explanation in the revised manuscript for all the details.

For the comments on the application range of proposed method, we agree that our proposed method approximates the nonlinear distortion as an additive noise having specific spectrum. Since nonlinear system is an extremely broad concept, we would not say our method could be applied to all nonlinear systems. This is also the reason that we added the “application range of PM NPR method” section in the article. However, the verified successful cases include the general Volterra model, the VCSEL and DFB laser cases dominated by the 2nd-order nonlinearity, the electrical driver case dominated by the 3rd-order memory nonlinearity, the optical coherent transmitter case having sine nonlinear function, the electrical DAC case having 3rd-, 5th-, 7th-, and 9th-order nonlinearities, the case of fiber nonlinear Kerr effect in both single span and multi-span transmissions. The application range is much wider than “cubic polynomial”. In addition, many nonlinear impairment sources in real-world communication applications including electrical circuits, the electrical DAC, optical devices (VCSEL, DFB laser), the optical coherent transmitter, and the optical fiber are verified. The proposed method is quite universal. The searching for exceptional cases is left for future research. Please refer to the section of “Application range of PM NPR method” in the revised manuscript.

2 - The technical description of the methodologies used is difficult to understand or is missing, leaving the readers to guess vaguely from the texts scattered over figures. For example, it is hard to understand Steps 1-3 in Fig. 2 from both text and the figure. What is the definition of the “resolution block”? What is the definition of the “difference between probability density functions”? There doesn’t seem to be any principle used in the paper that would make this “difference between probability density functions” decreases over iterations of Steps 1-3 (such as the gradient descent method), but why do the iterations lead to convergence? Why does the “local optimum” occur?

We add a supplementary file to describe the details of PM notch process. The “resolution block” is the spectrum resolution bandwidth. The “difference between probability density functions” is the difference between the PDF of the reference samples $PDF_{ref}(i)$ and the PDF of the signal after step 3 $PDF_{gen}(i)$. The

mathematical definition is $\int \sum_{Nj} PDF_{gen}(i) - PDF_{ref}(i)j$. The “perturbation” means the frequency points within the resolution bandwidth is multiplied by a random value $1+p*N(0,1)$, where a typical value of p is 0.05 and $N(0,1)$ is a standard-Gaussian sequence with a length of the number of frequency points within the resolution block.

Fig. S2 (figure 2 in the supplementary file) shows the PDF difference changes with iteration index when the perturbation is and is not used. If the perturbation is not used, the PDF difference decreases at first and saturates fast at a large PDF difference about 0.34. We compared the obtained sequences at iteration 80, 120, and 2400, and they are exactly same. The sequences could be found in the supplementary data set file. The PDF difference of 0.34 is not small enough. The iteration falls into the local optimum point.

If perturbation is used, the PDF difference decreases at first and fluctuates finally. The perturbation process keeps the overall spectrum beyond the resolution block but changes the details within the resolution block. From the time domain point of view, “the change within the resolution block” means that the signal sequence changes. In other words, “perturbation” generates another signal sequence but keeps the overall spectrum same. Many different signal sequences are hit during the iteration because the perturbation is random. Among those signal sequences, we could select a good one with enough small PDF difference and stop the iteration. For example, the good point in Fig. S2a has a PDF difference lower than 0.02.

Thus, the iteration is a “random searching” process and “convergence” is not necessary.

Please see the details in the supplementary file.

Fig. S2 PDF difference between the generated signal and reference samples. Random perturbation helps to find a good point with small PDF difference. (a) Results from the 1 iteration to 2400 iterations (b) Detailed results from 1 to 80 iterations.

We also add the explanation of PDF difference in the revised manuscript.

Here, the resolution block has the same concept as the resolution bandwidth of a spectrum. ... In the perturbation process, frequency components within each resolution block are multiplied by a set of random values. ... The difference between the PDF of the generated signal sequence and that of

reference samples are calculated as $PDF\ difference = \frac{1}{N} \sum_j PDF_{gen}(i) - PDF_{ref}(i)j$. N is the total number of bins in calculating the PDFs, and i is the bin index.

3 - Although the proposed method creates a desired probability density functions(PDF) of signal, the PDF is not made by independent and identically distributed signals but rather by correlated signals. This correlation does not exist in ordinary PAM signals, and can potentially cause artifacts when measuring NL distortion. To see this, note that when ordinary PAM signals are randomly permuted in time, the PSD does not change, but when the signals generated by the proposed method are randomly permuted in time, the PSD is likely to change. This means that the proposed method generates specially ordered signals in time, and since NL distortion can vary depending on the order of signal in time, the effect of the specially created order on the NL distortion should be analyzed.

We could see the impact of the special order in the case of multi-span fiber transmission. Please refer to the discussion of Fig. 9f and 9g in the revised manuscript. We show the key part in following.

To find the reason, the impact of notch bandwidth is investigated. Fig. 9f and 9g show the results of different notch bandwidth and the extrapolations at two notched frequency points in single- and multi-span transmissions. For single-span transmission, the NPR values are almost unchanged along with the notch bandwidth, while this phenomenon is different in multi-span transmission. The PM notch signal has a notch in the spectrum, so that it is correlated in time domain. In other words, the PM notch signal has a special order compared with the random signal in the real

communication. There is difference between the notched signal and the signal in real communication. This does not make sense in the single-span transmission, but makes sense in the multi-span transmission. The possible reason is the extremely long memory of CD in the multi-span

transmission.

4 - The use of a Chi-square distribution in some of the “equivalent additive noise models” has a physical basis, but shifting the Chi-square distribution to have zero mean is arbitrary and lacks a physical basis. In fact, if the exact mathematical model of an NL system is known, the PDF of NL distortion can be accurately determined through analysis or Monte-Carlo simulation in a similar way to the paper’s attempt with equations (3)-(4). However, the explanations surrounding equations (3)-(4) seem to be wrong. To be more specific, when a linear equalizer recovers $y_0 = c_1 * s_0 + z$, with z being NL distortion, equation (4) should first be rewritten in this form and the PDF of z should be characterized. Perhaps the dominant term of z would be product of two Gaussian random variables, whose PDF is a linear combination of two non-central Chi-square PDFs, not a single Chi-square PDF arbitrarily shifted to have a zero mean.

a) “Shifting the Chi-square distribution to have zero mean is arbitrary and lacks a physical basis”:

“zero-mean noise” is widely assumed in nonlinear system performance analysis [36, 39]. In a real-world implementation, this assumption is guaranteed by AC coupling in the receiver.

b) “The explanations surrounding equations (3)-(4) seem to be wrong”: In the equation $y_0 = c_1 * s_0 + z$, with z being NL distortion.

We do not clearly understand the comment. If “ $y_0 = c_1 * s_0 + z$ ” is the signal before receiver equalizer, “ z ” should include the linear ISI. If “ $y_0 = c_1 * s_0 + z$ ” is the signal after receiver equalizer, “ z ” is different from the additive noise in our model. In our model, the noise is added before equalizer. Since nonlinear distortion is not Gaussian, the PDF of the noise before and after equalization is different.

5 - For the “equivalent additive noise model” to be accurate, it needs an accurate PDF of the NL distortion. This requires identifying the mathematical model of the NL system under test, which is a very difficult task and may outweigh the benefits of the proposed method.

We agree that “finding the accurate PDF of the nonlinear distortion” is a possible way to estimate the nonlinear system performance. (Here, “nonlinear distortion” means equivalent nonlinear noise). Please notice that the one- dimensional PDF is not sufficient. The multi-dimensional PDF of nonlinear distortions at different times should be used. The receiver has a linear equalizer, and the system performance is determined by the noise after the equalizer and residual ISI. The noise after the equalizer is a linear combination of nonlinear distortions at different times. Since the PDF of nonlinear distortion is not Gaussian, only the spectrum is not sufficient, and the multi-dimensional PDF is necessary. This is even more difficult.

Due to the difficulty explained above, we do not select the approach of “finding the accurate PDF of the nonlinear distortion”. Our object is to estimate the nonlinear system performance accurately and practically but not to reproduce the complex nonlinear distortion. Consequently, our solution is to use the “zero-mean Chi-square” noise to approximate the very complex nonlinear distortion so that the nonlinear system performance could be estimated practically. The simulation and experiment results demonstrate that our approximation is accurate enough for long memory and lower order nonlinear systems. By the way, the orthogonal component has the accurate multi-dimensional PDF of the nonlinear distortion.

We add more illustration on the discussion of application range and PDF selection in revised manuscript. Please see the revised version.

Reviewer #2 (Remarks to the Author):-----

--

This paper proposes a probability-maintained (PM) NPR method and an equivalent additive noise model that can accurately estimate nonlinear system performance. The topic addressed in this paper is important and the manuscript is interesting. I have the following suggestions:

1. In the proposed PM NPR method, the spectrum needs to be adjusted, as shown in the Fig. 2a Step 2. An explanation about the 'perturbation in block' is required.

The “perturbation” means the frequency points within the resolution bandwidth is multiplied by a random value $1+p*N(0,1)$, where a typical value of p is 0.05 and $N(0,1)$ is a standard-Gaussian sequence with a length of the number of frequency points within the resolution block.

We add more detailed explanation about “perturbation” and the iteration process in the new supplementary file. Please refer to our response to the 2nd comment of reviewer #1 and the new supplementary file.

2. The notation 'NPR_{Simp}' represents the simple notch, what is the relation/difference between the simple notch and waveform-domain notch?

Unlike the 50-year-old conventional NPR method, the proposed PM notch signal is notched in the symbol-domain instead of the waveform-domain. The reason is illustrated in the 3rd graph of the section “Probability-Maintained NPR method”. All the “NPR_{Simp}” represents the simple notch at symbol domain in the article because PM-notch must be performed at symbol domain, and we need fair comparison between simple notch and PM-notch.

We revised contents as:

Unlike the 50-year-old conventional NPR method, the proposed PM notch signal is notched in the symbol-domain instead of the waveform-domain. “Waveform-domain notch” means the notch process occurs after digital pulse shaping, where as “symbol-domain notch” means the notch process occurs before digital pulse shaping. In the 50-year-old conventional NPR method, the notch process occurs before nonlinear device and it is “waveform-domain notch”. ... Thus, all the notch process include both PM notch and simple notch, are “symbol-domain notch” in following.

3. The equivalent nonlinear noise can be chosen following Gaussian noise or zero-mean Chi- square noise. How to determine the mean & covariance of the Gaussian noise, or the degrees of freedom of the zero-mean Chi- square noise? Please explain this point.

The means of all the additive noises are zero because of AC coupling in the receiver. The covariances of both the Gaussian noise and the zero-mean Chi-square noise are all zero at first. Then the NPR filter introduces the covariance determined by the noise spectrum which is obtained by PM-NPR method. The degree of freedom of the zero-mean Chi-square noise is 1 because the equivalent nonlinear noise is approximated as the square of a Gaussian random variable.

The revised contents:

The same-spectrum additive noise can be constructed by passing white X-distributed noise through the NPR filter. ... Zero-mean Chi-square is the PDF of a Chi-square distributed random variable with 1 degree of freedom minus its mean value. ... The mean of the Gaussian or Chi-square noise is zero since “zero-mean noise” is widely assumed in nonlinear system performance analysis^{36,39}. In real implementation, this assumption is guaranteed by alternative current (AC) coupling in the receiver. ... The NPR filters determine the power and covariance of additive noise.

Reviewer #3 (Remarks to the Author): -----

In my opinion, this paper is very well-written, with explanations that are very easy to understand, and figures that are well-thought-out. It is interesting to read.

However, there is an issue of how interesting and relevant it is. The technique of NPR is motivated by its (signal-processing) simplicity compared to the reference technique in the paper, correlated-orthogonal signal separation. However, the proposed technique PM-NPR is quite complicated signal-processing wise itself, which is clear from the description in the "Probability-maintained NPR method" section. The reference technique is not so complex or difficult to use at it is, and it is clearly better. And it also gives an antire spectrum as output, whereas the proposed PM-NPR only gives a single numerical value.

Further, the technique is quite ad-hoc, and the credibility is lost when it does not work for all cases (e.g. for fiber nonlinearity), and the authors have no good explanation why. The Chi-Square noise distribution is also weakly motivated, really only through simulations and experiments, and it does not always work.

Thus, the paper has a bit of an "unfinished" feeling, the technique is a bit ad- hoc, and it is questionable if it is worth to study it at all, when the reference technique is quite useful directly.

Conclusion: Interesting paper, but it has quite significant shortcomings. At this point, I am inclined to recommend rejection.

1. “the proposed technique PM-NPR is quite complicated signal-processing wise itself”

We agree that the digital signal processing of PM-notch is more complicated than that of conventional simple notch NPR and orthogonal decomposition. However, the PM-notch process is nothing but to find a special symbol sequence which has a notch in the spectrum and keeps the probability. Such a process is performed only ONCE in the design stage of the test symbol sequence. The test symbol sequence obtained by one person could be used by any other person at any time. This is also the reason we disclose the PM-notch sequence in the article.

In addition, the hardware requirement of PM-notch is even lower than that of conventional NPR. The PM-notch sequence is just a symbol sequence with a special order. PM notch signal could be generated by transmitter itself, whereas the conventional NPR needs another instrument in the transmitter side.

Please refer to the new “Complexity and implementation considerations” part in the discussion section

2. “The reference technique is not so complex or difficult to use at it is, and it is clearly better. And it also gives an entire spectrum as output, whereas the proposed PM-NPR only gives a single numerical value.”

We think the reference technique here means the conventional orthogonal decomposition method. If it refers to conventional NPR method, it is clear that the conventional NPR cannot be used for non-Gaussian stimuli. We agree that the orthogonal decomposition is easy to understand, and it could obtain the entire spectrum. However, the real implementation of the orthogonal decomposition is prohibitively challenging and tricky because it needs the accurate measurement and comparison of the input signal and the output signal [33]. The reasons are illustrated in following.

- a) The orthogonal decomposition method needs very expensive measurement instruments, such as the high speed arbitrary waveform generator and the digital storage oscilloscope. The price of such instrument is several times higher than the spectrum analyzer used in the PM notch method.
- b) The orthogonal component $y_o(t) = y(t) - y_c(t)$ is a small difference between two large signals. A small measurement error of a large signal caused a significant error in the orthogonal component. Supposing the NPR is -20 dB, then the power of the $y(t)$ will be 100 times larger than that of the $y_o(t)$. If the measurement error of the $y(t)$ is 1%, the

- error of the $y_o(t)$ is comparable with the $y_o(t)$ itself.
- c) Some nonlinear device has different types of input and output signals. For example, in the case of HB-CDM which is the main nonlinear device in the optical coherent transmitter, the input signal is an electrical analog signal, whereas the output signal is an optical signal. Comparison of electrical signal and optical signal is very challenging and tricky³³.
 - d) For the cases of fiber Kerr effect, there are the laser phase noise, the amplifier spontaneous emission noise, and the polarization uncertainty in the real-world transmission. Those unknown noises also cause significant errors in orthogonal decomposition.

The proposed PM NPR method only needs the transmitter to send a predefined sequence and a spectrum analyzer to measure the output spectrum. All above difficulties are overcome.

Please refer to the new “Complexity and implementation considerations” part of the discussion section in the revised manuscript.

- 3. Further, the technique is quite ad-hoc, and the credibility is lost when it does not work for all cases (e.g. for fiber nonlinearity). The Chi-Square noise distribution is also weakly motivated.

Please refer to our response to the 1st, 4th and 5th comment of reviewer #1.

Finally, we revised a typo in the last sentence of “Same-spectrum model construction” in the “Method” section:

According to our experimental results, the low resolution of these filter responses, for example, 150 MHz of OSA in the optical B2B system, is acceptable.

REVIEWERS' COMMENTS:

Reviewer #3 (Remarks to the Author):

The authors have written a very good rebuttal, which I am mainly fine with (I am reviewer 3). The motivation of why the technique is valuable is convincing. I am not as convinced about the use of the zero-mean chi-squared distribution, it feels very ad-hoc and uncertain still, but in general the paper is improved and explanations provided so that the paper is acceptable now.

In response to Reviewer 1's Comments:

I have closely read the answers to the reviewer comments from the authors, and I must say that they have answered in a very good way, and convinced me that the paper is worth publishing. I have no further comments on the current version of the manuscript.